# Global risk pooling mitigates financial risk from drought in hydropower-dependent countries

Rosa Isabella Cuppari [1,2] ✉, Tamlin M. Pavelsky [3] & Gregory W. Characklis [1,2]

More than 50 countries rely on hydropower for over 25% of their electricity generation, making them vulnerable to drought and resulting revenue losses. Governments can offset financial losses for publicly-owned hydropower generators, but this can create fiscal pressures and lead to negative consequences, such as lower bond ratings. Index-based financial instruments, used to manage weather-related risk, offer an alternative, though data collection and index design are challenging. Using remotely sensed hydrometeorological data, we develop index insurance contracts to manage drought-related financial risk for hydropower-dependent countries. Low correlations in drought across these countries allow cost reductions when risks are pooled. Pooling the contracts yields average savings of 54% compared to individual risk management via reserves. These findings indicate that pooled index insurance can strengthen financial resilience in countries dependent on hydropower and support governments in mitigating drought-related economic risks.

Hydropower accounts for the majority of global renewable energy, a share expected to grow through 2030 as countries attempt to meet net zero emissions targets and energy-related Sustainable Development Goals (SDGs)[1]. For 56 countries, however, hydropower already provides more than 25% of domestic electricity generation[2] and, for some, hydropower sales are also an important source of government revenue (Fig. 1). Yet hydropower's advantages are undermined by its vulnerability to hydrometeorological variability - the first half of 2023 alone saw an 8.5% decline in global hydropower generation due to droughts[3], though individual countries experienced far larger drops (e.g., China – 20%[4]). Corresponding reductions in revenues of this scale may impact a hydropower supplier's financial stability, as hydroelectric dams require large, upfront capital expenditures that are usually repaid at constant rates. Sufficiently large drops in hydropower revenue can lead to financial losses, undermining a supplier's ability to meet these fixed debt service obligations.

For the governments that rely on hydropower, reductions in generation and the commensurate loss of revenues may yield budget deficits[5], which are most difficult to manage for low and lower-middle income countries (LICs/LMICs). The need to purchase more expensive, compensatory sources of electricity to meet firm contractual obligations can compound these revenue losses by simultaneously increasing costs[6]. And, in many regions, droughts and heatwaves are at least somewhat correlated[7], further exacerbating the impacts of reduced generation as heatwaves lead to increased electricity demand (e.g., for air conditioning). Such variable revenues and the increased potential for financial losses at the national level can lead to sovereign credit rating downgrades, driving interest rates on government borrowing higher, and thus increasing the costs thereof. This is particularly important for LICs/LMICs, of which less than 5% hold investment-grade credit ratings[8]. For example, Fitch credit rating agency (Fitch Ratings) noted in its analysis of Zambia, which uses hydropower to generate over 80% of its electricity[9], that the country would face substantial costs to import electricity during a drought[10]. Fitch expected this increase would be large enough to adversely impact the national budget, negatively influencing the country's credit rating.

For these reasons, hydropower suppliers at all scales have developed various strategies to manage their hydrometeorological financial

[1]Department of Environmental Sciences and Engineering, University of North Carolina at Chapel Hill, Chapel Hill, NC, USA. [2]Institute for Risk Management and Insurance Innovation, University of North Carolina at Chapel Hill, Chapel Hill, NC, USA. [3]Department of Earth, Marine and Environmental Sciences, University of North Carolina at Chapel Hill, Chapel Hill, NC, USA. ✉e-mail: rosa.cuppari@gmail.com

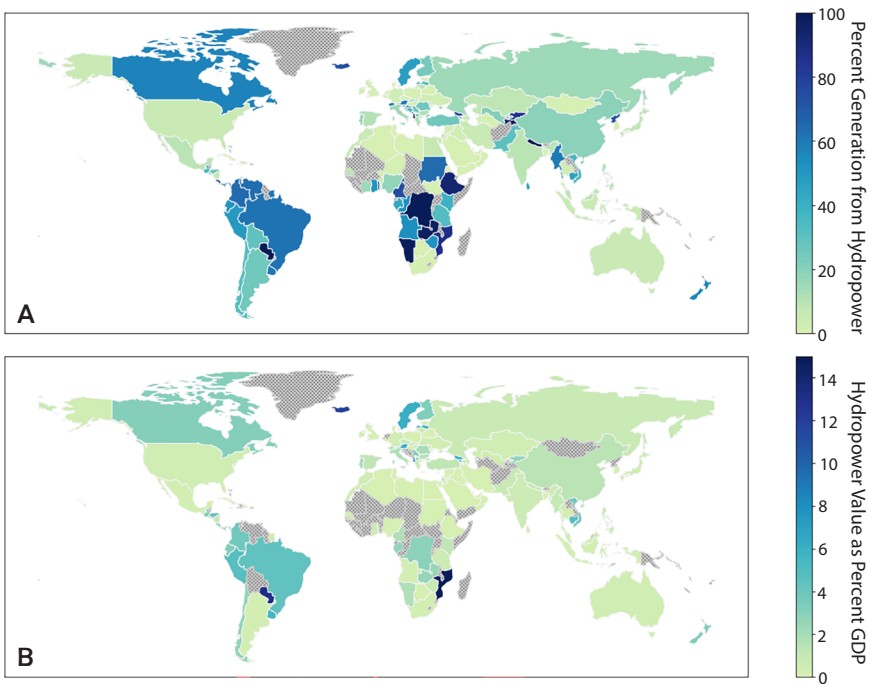

**Fig. 1 | National-level dependence on hydropower. A** Percent of total generation from hydropower in 2015[2]. **B** Hydropower sales as a percent of GDP. Sales as percent of GDP are calculated using 2022 generation[67], GDP[102], and tariffs[103], where available. Grey, cross-hatched countries do not have generation and/or tariff information available.

risk (Fig. 2). A traditional risk reduction strategy might involve building larger reservoirs to store more water for use during drought, or diversifying the energy mix. Such approaches typically require large capital outlays for capacity which will be seldom used and which may ultimately fail to provide the expected level of risk management under uncertain future climate conditions. Another category of management techniques is risk retention, which often involves holding cash reserves to protect against low revenue and/or high-cost periods (self-insurance). However, reserves must be maintained in a relatively liquid form to be available for unpredictable events, and therefore earn little return. Given the sums necessary to protect against the most extreme events, these low returns can lead to prohibitively high opportunity costs associated with holding large reserves for long periods—the foregone returns when holding funds as opposed to investing them - especially in LICs/LMICs where capital is scarce.

Risk transfer techniques are often better suited for managing larger, more infrequent losses[11]. This approach typically involves paying a third party (or parties) to take on the risk of losses in exchange for payment. Transfer is often conducted through financial contracts, such as index insurance, whose payouts are triggered when a threshold of some pre-established metric correlated with financial losses ("the index") is crossed. Such index-based contracts can link weather-based losses to payouts intended to compensate for them. In comparison to traditional, indemnity-based insurance, which involves the filing and individual adjudication of every loss claim, index insurance can provide faster and less contentious payouts. The pre-set trigger and payment structure can also reduce administrative costs for the insurer and, therefore the buyer. Contracts using weather-based indices can be compared to weather-based derivatives[12–14], which emerged in the 2000s to mitigate risk for power utilities and generators experiencing fluctuations in electricity use (and revenues) based on daily temperatures.

If the risks in question are spatially uncorrelated or negatively correlated, insurance and other risk transfer tools can be pooled across regions. Establishment of risk pools consisting of parties with uncorrelated losses can reduce the reserves required to offset a given level of losses, relative to each pool member individually maintaining

adequate reserves, thereby lowering the costs of insurance[11,15,16]. Several regional and global risk pools take advantage of this strategy, including the Caribbean Catastrophe Risk Insurance Facility (CCRIF), the Pacific Catastrophe Risk Insurance Facility (PCRIF), and African Risk Capacity, each of which provide insurance to countries as a means to mitigate household-level property and agricultural losses resulting from hurricanes, tropical storms, and/or droughts. Empirical evidence from these organizations suggests that pooling hydrometerological financial risks reduces the cost of insurance by 27% to 65%[11,17,18], values that are dependent on the level of correlation in losses across pool members. However, despite the advantages to pooling, such an approach has not yet been applied to insuring the hydropower sector.

One challenge to developing risk pooling strategies at the global level is data availability, as the design of robust risk transfer instruments and risk pools requires historical data to characterize the probability and severity of droughts, and limited on-site historical data are available in many hydropower-dependent countries[19]. In previous works, satellite-based measurements of vegetation, elevation, temperature, and precipitation have been used to characterize the severity of drought[20,21]; crop growth and yields[22–24]; land use/land cover change[25,26]; flooding[27]; and streamflow[28,29], as well as to generate seasonal streamflow forecasts for hydropower generation[30]. Such data can be used to develop multivariate indices that correlate with financial losses from reduced hydropower generation, facilitating the development of new risk transfer tools such as a risk pool to combat drought-related losses for hydropower.

We use increasingly available remotely sensed data to do so, designing index insurance contracts which form the basis of a global risk pool that can cost-effectively manage drought-related financial risk in countries that rely on hydropower. First, country-level index contracts are designed using remotely sensed data, which while limited, provide greater temporal and spatial coverage than alternative data, such as streamflow reanalysis datasets (e.g.,[31]). The contracts are then individually evaluated and priced. Second, these contracts are pooled, with each country experiencing savings on the cost of the insurance contract based on the degree to which their inclusion in the pool influences the magnitude of, and level of correlation in, payouts.

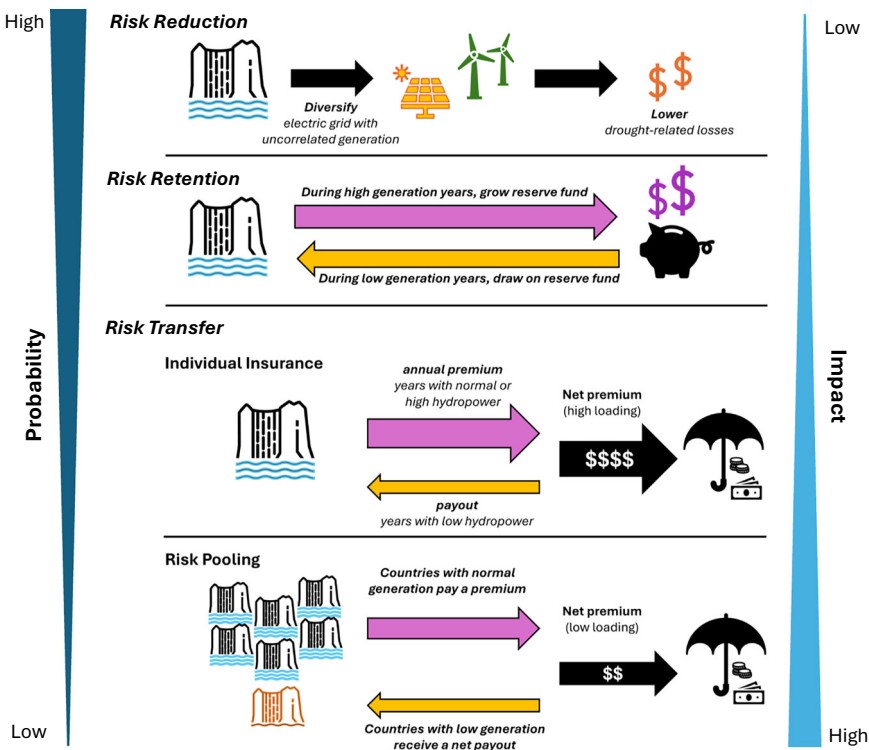

**Fig. 2 | Comparison of risk management strategies.** The risk of high probability, low impact events is typically managed using risk reduction or risk retention. Risk transfer is best suited for low probability, high impact events.

The costs of the individual and pooled insurance contracts are compared to each other, and to the opportunity cost of self-insuring by holding liquid reserves. Results suggest that hydrometeorological data relevant to both water supply and electricity demand (i.e., precipitation, temperature, snow cover extent, and vegetation indices) can be used to develop index insurance contracts that can be pooled to provide an average savings of 54% in comparison to each pool member holding individual reserves. Our findings are relevant to both governments seeking to reduce their exposure to hydrometeorological variability and to international finance institutions interested in developing effective risk management tools. They suggest an opportunity to stabilize hydropower-dependent countries' energy finances such that additional resources can be used to invest in alternative generation capacity or other development priorities. This framework may be particularly useful for LICs and LMICs that are both subject to substantial financial risk and have limited resources to independently manage it.

## Results

### Index insurance contracts

Monthly data for four hydrometeorological conditions are used as inputs to indices that correlate with annual national hydropower generation (Table 1). Precipitation, retrieved from the Integrated Multi-satellitE Retrievals for the Global Precipitation Measurement Mission (IMERG), is a proxy for water supply[32]. Snow cover extent (Normalized Difference Snow Index, NDSI) and vegetation indices (Normalized Difference Vegetation Index, NDVI and Enhanced Vegetation Index, EVI) similarly provide information on water availability, with the former most important in basins where streamflow is driven by snowpack. Available snow and vegetation index data[33,34] are produced from observations of the Terra Moderate Resolution Imaging Spectro-radiometer (MODIS). Finally, land surface temperature (LST) data from MODIS[35] provides insight into both water supply, as it is influenced by on-the-ground moisture, and electricity demand, as electricity use is linked to high temperatures[36].

The multivariate indices, designed using linear regressions for each of the 56 hydropower-reliant countries, form the basis of the index insurance contracts. In order to be effective, the indices on which these contracts are based must be reasonably correlated with hydropower generation and revenues. Of the countries considered, 15 have indices with $r^2$ values ≥0.35 (see Fig. 3) and trigger at least one payout during the 21-year historical timeseries (2002–2023). This threshold corresponds to the low end of commercially available index insurance contracts[37–39]. Notably, 67% of the 15 countries in our pool have three or fewer dams whose 'Main' or 'Major' purpose is to generate hydroelectricity (as categorized in the Global Reservoir and Dam Database (GRaND) database). The remaining five participants (Norway, Austria, Albania, Croatia, and Georgia) have a greater number of dams, though only Norway and Austria have dams on more than 5 rivers, as designated by the GRaND database. This suggests that future efforts to form risk pools might initially focus on countries with concentrated hydroelectric facilities. The minimum $r^2$ threshold for the fitted model limits basis risk, which represents the ability of an index to trigger payouts when losses occur and at an appropriate amount. Additional leave-one-out cross-validation provides further confidence in these indices, as all indices have mean absolute errors of ≤0.11 across a range of training and testing set combinations, relative to the capacity factor range of 0–1. The Normalized Nash-Sutcliffe Efficiency for each index lies between 0.57 and 0.74, indicating adequate fit. Supplementary Information Table 2 (Section "Additional Statistical Tests") provides results for additional statistical tests. The indices are also limited to four or fewer indicator variables, making them relatively transparent to buyers, a factor important for increasing insurance uptake[40]. The remainder of this analysis exclusively focuses on the 15 countries whose indices meet these criteria.

Ten indices utilize data aggregated at the subbasin level, which takes into consideration the outsized impact of large dams clustered in a single subbasin and/or disparate climatic zones across a country. For example, Chile's hydropower capacity is concentrated in a single region although the country spans nearly 40 degrees of latitudes. Five

## Table 1 | Datasets (a) and aggregations (b) used in index design

**(a) Datasets**

| Variable | Source | Definition | Importance |
|---|---|---|---|
| Precipitation | Integrated Multi-satellitE Retrievals for the Global Precipitation Measurement Mission (IMERG)[32] | Integrated precipitation data collected from passive microwave estimates, microwave-calibrated infrared estimates, and precipitation gauges | Proxy for water supply |
| Normalized Difference Snow Index | Terra MODIS[33] | The normalized difference of green and shortwave infrared wavelengths | Proxy for water supply |
| Normalized Difference Vegetation Index | Terra MODIS[34] | The normalized difference between the red and near-infrared wavelengths | Proxy for water supply |
| Enhanced Vegetation Index | Terra MODIS[34] | The normalized difference between the red and near-infrared wavelengths, corrected for atmospheric conditions and canopy background noise | Proxy for water supply |
| Land surface temperature | Terra MODIS[35] | The emissivity and "brightness" temperature of the surface in Kelvins | Proxy for electricity demand |
| Hydropower capacity | International Renewable Energy Agency[68] | National hydropower capacity (MW) | Used to account for changing hydroelectric capacity over time |
| Watershed boundaries | HydroBASINS[66] | Subbasin level delineation using the location where two river branches meet, given a minimum upstream area of 100 km$^2$ | Used for spatial aggregation of remotely sensed data |
| Hydropower generation | US Energy Information Administration[67] | Observed national-level generation (TWh) at the annual time step | Outcome of interest |

**(b) Spatial and temporal aggregation**

| Name | Abbreviation | Definition |
|---|---|---|
| Winter season | Winter | Average value for December, January, and February |
| Spring season | Spring | Average value for March, April, and May |
| Summer season | Summer | Average value for June, July, and August |
| Fall season | Fall | Average value for September, October, and November |
| HYBAS 4 | _4 | Average value across HYBAS 4 level subbasin with the greatest concentration of dams |

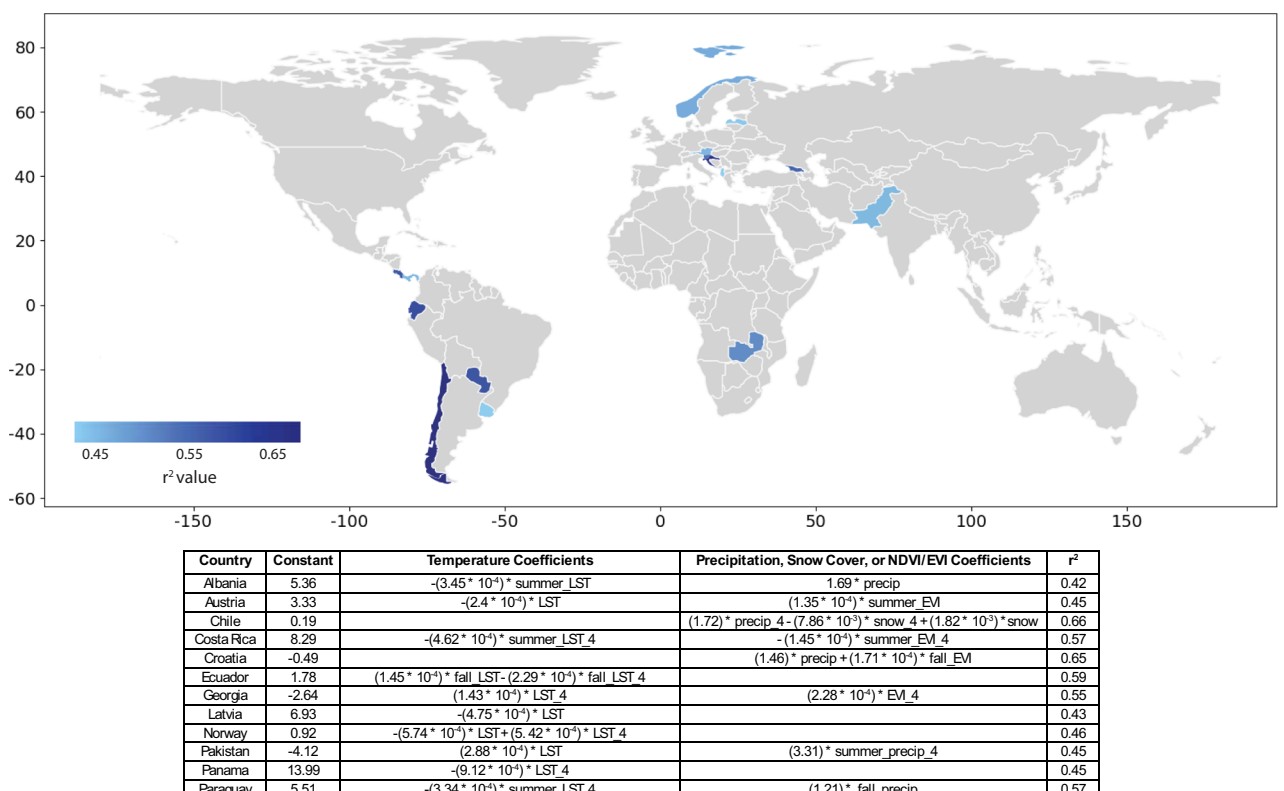

| Country | Constant | Temperature Coefficients | Precipitation, Snow Cover, or NDVI/EVI Coefficients | $r^2$ |
|---|---|---|---|---|
| Albania | 5.36 | $-(3.45 * 10^{-4}) *$ summer_LST | $1.69 *$ precip | 0.42 |
| Austria | 3.33 | $-(2.4 * 10^{-4}) *$ LST | $(1.35 * 10^{-4}) *$ summer_EVI | 0.45 |
| Chile | 0.19 | | $(1.72) *$ precip_4 $- (7.86 * 10^{-3}) *$ snow_4 $+ (1.82 * 10^{-3}) *$ snow | 0.66 |
| Costa Rica | 8.29 | $-(4.62 * 10^{-4}) *$ summer_LST_4 | $- (1.45 * 10^{-4}) *$ summer_EVI_4 | 0.57 |
| Croatia | -0.49 | | $(1.46) *$ precip $+ (1.71 * 10^{-4}) *$ fall_EVI | 0.65 |
| Ecuador | 1.78 | $(1.45 * 10^{-4}) *$ fall_LST $- (2.29 * 10^{-4}) *$ fall_LST_4 | | 0.59 |
| Georgia | -2.64 | $(1.43 * 10^{-4}) *$ LST_4 | $(2.28 * 10^{-4}) *$ EVI_4 | 0.55 |
| Latvia | 6.93 | $-(4.75 * 10^{-4}) *$ LST | | 0.43 |
| Norway | 0.92 | $-(5.74 * 10^{-4}) *$ LST $+ (5.42 * 10^{-4}) *$ LST_4 | | 0.46 |
| Pakistan | -4.12 | $(2.88 * 10^{-4}) *$ LST | $(3.31) *$ summer_precip_4 | 0.45 |
| Panama | 13.99 | $-(9.12 * 10^{-4}) *$ LST_4 | | 0.45 |
| Paraguay | 5.51 | $-(3.34 * 10^{-4}) *$ summer_LST_4 | $(1.21) *$ fall_precip | 0.57 |
| Slovenia | 0.17 | | $(4.95 * 10^{-3}) *$ summer_snow_4 $+ (0.99) *$ summer_precip_4 | 0.53 |
| Uruguay | 28.31 | $-(1.83 * 10^{-3}) *$ LST | $-(2.09 * 10^{-4}) *$ fall_EVI | 0.42 |
| Zambia | 0.30 | $(6.77 * 10^{-4}) *$ spring_LST_4 $- (6.53 * 10^{-4}) *$ LST_4 | $(0.70) *$ precip_4 | 0.50 |

**Fig. 3 | Performance of indices along with regression formula.** Variables are averaged at the country level or, when ending with "_4", at the subbasin scale, using the subbasin with the greatest number of hydroelectric dams. Seasons reflect Northern Hemisphere months (e.g., winter indicates December-February). Two-tailed p-values are listed in Supplementary Information Table 1.

indices exclusively use annually aggregated hydrometeorologic conditions as predictor variables, while the remainder have at least one seasonally aggregated predictor (e.g., average fall temperature). Variables aggregated at different seasonal time steps provide information related to the timing of hydropower generation—an important consideration given that periods of high supply may not align with periods of high demand. Temperature and precipitation—variables that act as proxies for both electricity demand and hydropower generation—dominated index inputs. Snow cover extent and EVI—both indicators of water supply and therefore hydropower generation—were also included. Less than half of the indices rely on either temperature or water supply indicators alone to predict hydropower generation.

Insurance contracts are designed to pay out when indices fall below the twentieth percentile of a country's historical capacity factors. Payouts are capped at the worst 0.5% of losses, also referred to as the 99.5% Value at Risk (VaR). The expected annual payout for the individual contracts is $20 million (cumulatively $301 million). This value, however, varies widely for individual countries. Slovenia (1172 MW), the country with the least installed hydropower capacity in the pool, has an expected annual payout of $4.3 million, while Chile has far larger hydropower capacity (6934 MW) and an expected annual payout of $42 million.

Contract premiums are a function of these expected payouts and a loading factor, which represents the portion of the premium in excess of the expected payouts and accounts for the costs of contract design, administration, and the insurer's return. The loading coarsely represents the cost of insurance (ignoring the time value of money), as it is the fraction of the premium that does not statistically return to the buyer over the long-term. Here, we use the Wang Transform[41] to price contracts, which applies a risk adjustment factor ($\lambda$) to modify the

distribution of payouts in order to more heavily weigh infrequent, but high magnitude payouts, such that the mean of the modified distribution represents the expected payout and loading (see "Methods", Section "Index insurance").

Large variability in payouts, along with their magnitudes, is reflected in individual country premiums, which vary between $1.7 million for Latvia and $107 million for Norway, as well as the loading (see Table 2). While the range of premiums reflects the magnitude of payouts required for different countries, a function of absolute hydropower capacity and generation, differences in loading (37–64% in our sample) attest to the variability in payouts for each country. Austria and Croatia, for example, experience low levels of variability, as measured by the ratio of the standard deviation to mean payout value (coefficient of variation), and correspondingly have lower levels of loading than other countries in the pool (37% and 39%, respectively). In contrast, Pakistan and Latvia both exhibit high levels of variability and experience higher proportionate loading (64%).

The cost-effectiveness of the individual contracts can be compared to the opportunity cost of self-insurance via a reserve fund, i.e., foregone returns from holding liquid funds large enough to cover losses equivalent to the maximum insurance payouts (see "Methods", "Index insurance"). As seen in Fig. 4A, risk transfer via individually purchased insurance provides the same level of risk mitigation as reserves at 65% of the average cost. The benefits of purchasing individual insurance versus holding reserves vary by country with some countries experiencing cost savings of over 50% (Table 2 and Fig. 4B). However, self-insurance can be more cost-effective than individually purchased insurance. In this pool, the costs of risk management for Croatia increases when purchasing individual insurance. That said, the relative cost-advantage of purchasing insurance is contingent on the

**Table 2 | Hydropower capacity (2020) and premiums for countries under individual insurance contracts and within the risk pool**

| Country | Hydropower generation capacity (MW) | Expected pay-out ($M) | Opportunity cost of reserves ($M) | Individual policy loading ($M) | Pooled policy loading ($M) | Loading savings (%) |
|---|---|---|---|---|---|---|
| Albania | 2387 | 4.6 | 8.3 | 3 | 1.8 | 40 |
| Austria | 8903 | 32.5 | 17.3 | 11.9 | 11.0 | 8 |
| Chile | 6934 | 42.3 | 20.6 | 19.6 | 15.9 | 19 |
| Costa Rica | 2332 | 10.7 | 5.3 | 5.0 | 3.2 | 36 |
| Croatia | 1924 | 16.3 | 5.4 | 6.3 | 3.2 | 49 |
| Ecuador | 5098 | 20.1 | 13.8 | 9.1 | 4.1 | 55 |
| Georgia | 3323 | 12.9 | 6.7 | 5.7 | 2.7 | 53 |
| Latvia | 1586 | 1.1 | 2.7 | 0.7 | 0.04 | 94 |
| Norway | 32,285 | 70.4 | 47.1 | 37.0 | 33.0 | 11 |
| Pakistan | 10,045 | 18.4 | 36.3 | 11.8 | 6.4 | 45 |
| Panama | 1810 | 6.9 | 5.6 | 3.8 | 2.0 | 47 |
| Paraguay | 8824 | 30.3 | 35.4 | 16.3 | 13.1 | 19 |
| Slovenia | 1172 | 4.3 | 3.3 | 2.4 | 1.9 | 20 |
| Uruguay | 1538 | 12.5 | 8.2 | 5.7 | 2.9 | 49 |
| Zambia | 2400 | 17.8 | 8.1 | 7.5 | 2.8 | 62 |

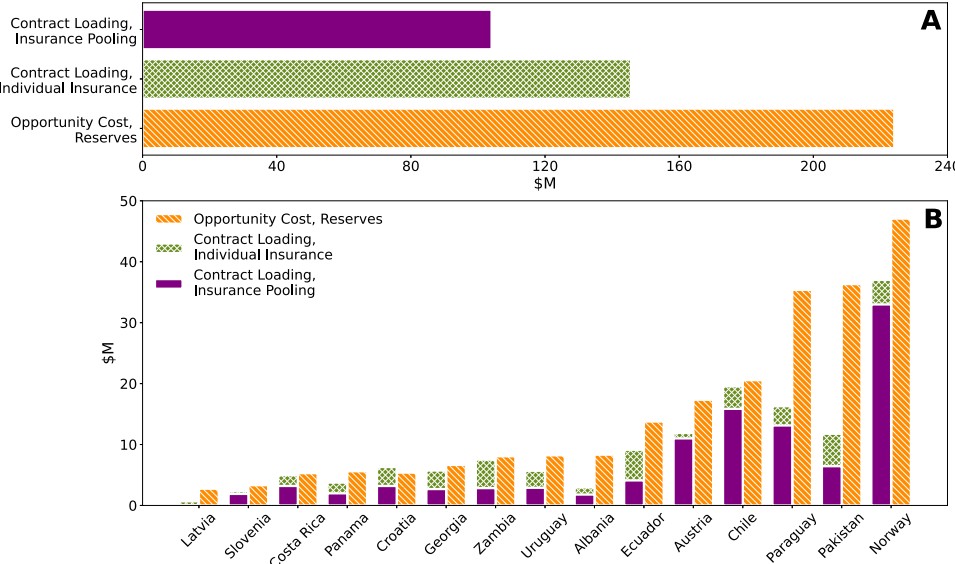

**Fig. 4 | Relative cost of holding reserves and purchasing insurance, individually or as part of the pool. A** The aggregate annual opportunity costs of holding reserves up to the maximum insurance payout compared to the loading on insurance premiums. **B** The per country opportunity cost compared to loadings. Bars representing loading are additive.

rate of returns. Figure 5 (Section "Index insurance") shows historical rates of return under which self-insurance is more or less attractive than purchasing individual insurance; higher rates of return generate higher opportunity costs, making insurance more attractive.

**Savings from pooling reserves and insurance**

The risks of the individual countries and the individual insurance contracts can be pooled, with premiums modified to reflect the new, aggregate risk of the pool. The impact of pooling insurance contracts mirrors the reductions in costs when reserves are pooled instead of individually held.

This is because as the number of pool participants grows, the coefficient of variation associated with the aggregate payout falls, even as the overall, annual expected payouts increase. This phenomenon can be attributed to the relatively low levels of correlation between

each country's hydropower generation, and drives the difference between the reserves necessary to compensate for the 99.5% VaR of the pool and the sum of the reserves countries would need to individually hold in order to independently manage their risk. Figure 6 shows the reduction in reserves achievable with multiple countries in the pool, whose cumulative reserves across all potential participants, if held individually, would amount to $4 billion. Instead, when reserves are jointly held within a pool, the necessary amount falls by -33% ($1.3 billion) to $2.7 billion. This corresponds to the smaller reserves that an insurer would need to maintain as well, and in a competitive market lower costs for an insurer should lead to lower loading for purchasers[42].

The reduction in necessary reserves manifests itself as the difference between premiums for individually insured countries versus premiums charged when they are members of the pool. Were the 15 countries analyzed in this work to purchase individual insurance

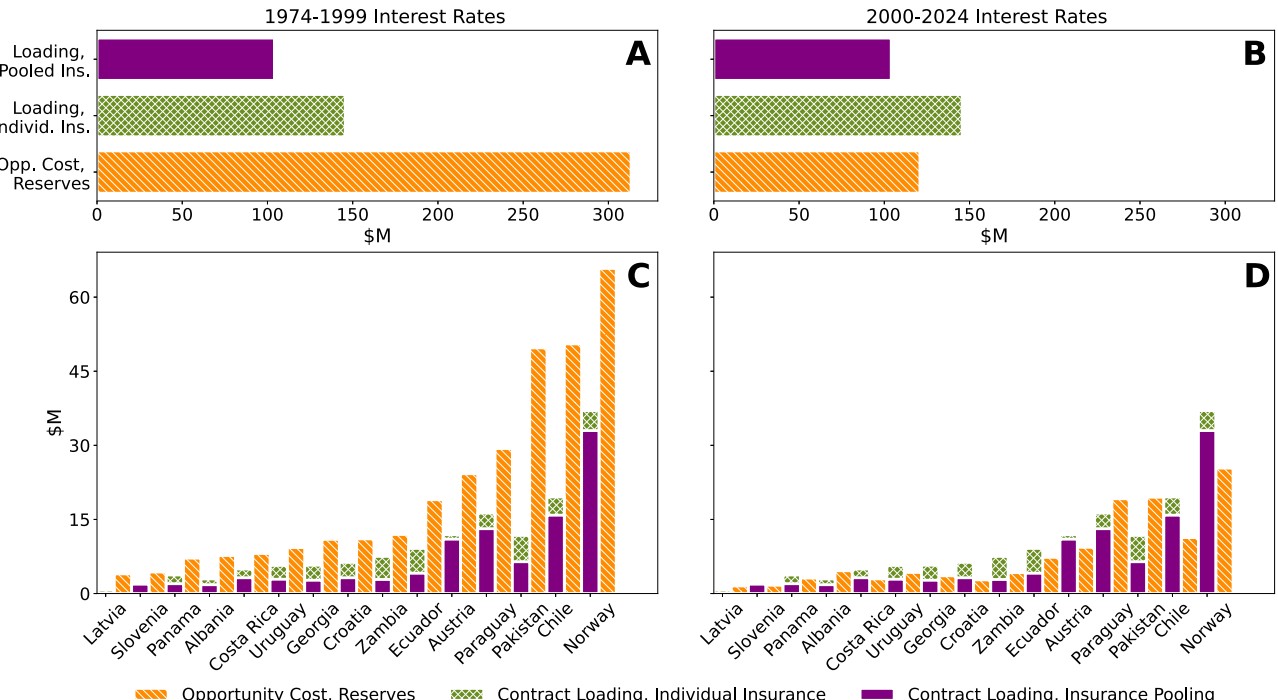

**Fig. 5 | Comparison of individual insurance contract loading, pooled insurance contract loading, and reserve opportunity costs at low and high interest rates.** **A**, **B** Aggregate loadings and opportunity cost for 15 countries in the pool while **C**, **D** show per country contract loadings and opportunity cost. Left column (**A**, **C**) uses average Treasury bond yields from 1974 to 1999 and the right column uses average yields from 2000 to 2024 (**B**, **D**).

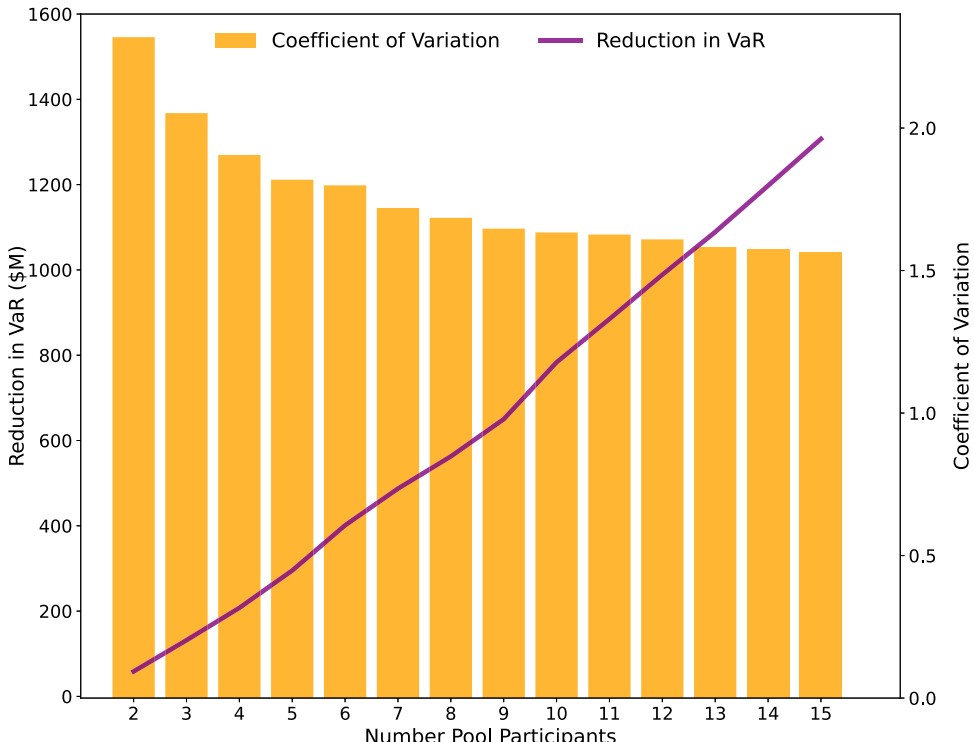

**Fig. 6 | Changes in the 99.5% Value at Risk and coefficient of variation for payouts as pool size increases.** Pooling reserves reduces the levels needed to meet the 99.5% VaR of participating countries when compared to these countries individually holding reserves. Increasing the number of participants in a pool leads to greater savings. Values by pool size are calculated as the average savings across different combinations of pool participants.

contracts, the sum of their premiums would equal $447 million with an average loading of 50% ($146 million total). In contrast, the overall premium for the risk pool is $405 million, with an average loading of 29% ($104 million total). This overall annual saving of $42 million is distributed across pool members by calculating each participant's Shapley value (Methods "Global risk pool"), which captures the changes in contract loading attributable to each country's participation in the pool.

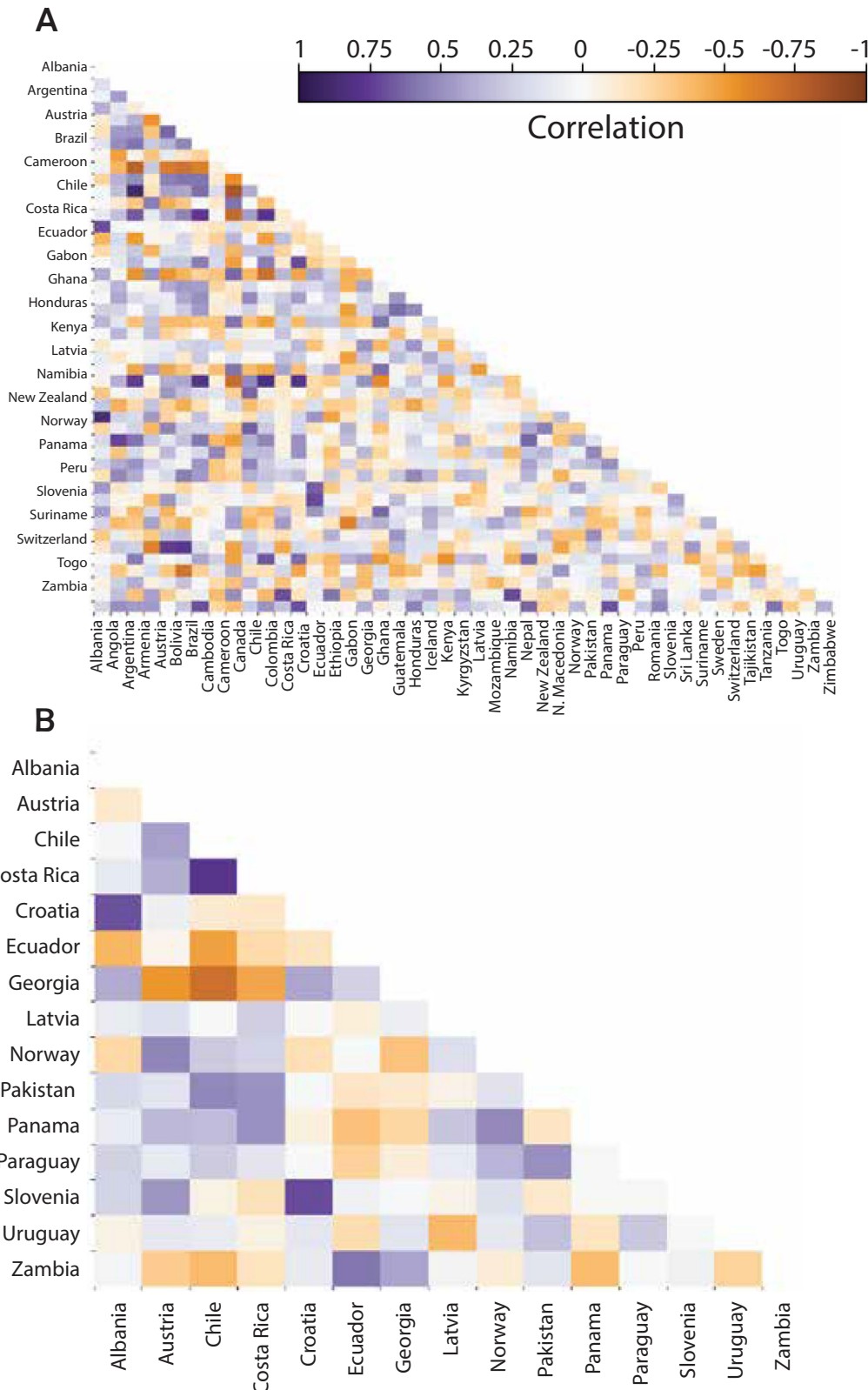

Fig. 7 | **Correlation of national level capacity factors.** **A** Hydropower-dependent countries globally. **B** Countries within the pool. The pool comprises all countries for whom indices with $r^2 \geq 0.35$ and less than four indicator variables could be developed.

Using the Shapley value to allocate savings from participation in the pool with all 15 participants, individual participants see reductions in loading between 8% and 94% (see Table 2 and Fig. 4B), with the greatest savings going to countries whose payouts are uncorrelated with others

in the pool (e.g., Latvia and Zambia) and therefore bring more value in terms of diversification (see Fig. 7B). The smallest reductions in loading occur for the countries with the highest payouts or those most correlated to the pool, e.g., Norway and Austria. The reduction in contract

loading from establishing a risk pool are such that insurance is more cost-effective than holding a reserve fund for all countries in the pool and remains cost-effective for most countries even as the opportunity cost of holding reserves falls (Fig. 5, Section "Index insurance")

## Discussion

In 2015, the United Nations (UN) adopted two historic global frameworks: the SDGs, a set of global goals for achieving a sustainable and prosperous future, and the Sendai Framework for Disaster Risk Reduction, a blueprint for increasing countries' resilience to natural disasters[43]. These complementary initiatives established priorities for investing in sustainable infrastructure, enhancing resilience to climate and financial shocks, and adapting to an uncertain climate future. In so doing, the initiatives gave rise to a new, innate tension – increasing renewable energy penetration to reduce emissions while also managing the impacts of extreme weather events, which can have notable impacts on renewable energy sources like solar, wind, and hydropower.

The Sendai Framework also identified a potential solution, namely national-level risk transfer instruments similar to the index insurance contracts developed in this work. Such contracts provide a means to transparently calculate and expedite payments to countries experiencing extreme events, compensating for losses in impacted sectors and potentially contributing to financing of risk reduction measures. Additional calls have followed since 2015, including through a 2017 report from the World Bank to the Group of 20 (G20) countries which recommended scaling up catastrophe risk pools from the local to national governance levels[11]. Countries are heeding these recommendations, joining regional risk pools like CCRIF and PCRIF or purchasing index-based instruments to manage the risk of natural hazards such as droughts, earthquakes, floods, tsunamis, and windstorms[44,45]. Partnerships between a large international finance institution (e.g., the World Bank) and large (re)insurers (e.g., Swiss Re) have enabled the development of these tools, taking advantage of the financial stability of the former in combination with the access to capital and well-diversified portfolio of the latter. This type of partnership has been applied to insure Uruguay's hydropower sector against the risk of drought[5], as well as to insure against multiple natural hazards in Mexico, via The Fund for Natural Disasters of Mexico (FONDEN)[46,47]. At a broader level, national- and regional-level risk pools are exploring joint reinsurance facilities to further diversify their risk[48], reducing the levels of needed reserves and therefore lowering the costs of risk management for pool participants. Nonetheless, greater potential remains for applying risk pooling to other sectors, in particular renewable energy. Our analysis addresses this gap for hydropower, assessing the potential for a global risk pool to manage hydrometeorological financial risk and contributing to efforts to expand the number of risk pools—all of which can make insurance more practical in small and/or low-income countries.

Indeed, as is the case for other risk pools, we find that index insurance can mitigate the financial impacts of severe, drought-related hydropower shortfalls and that pooling risks across countries can substantially reduce premiums for all pool members. Other renewable energy sources whose generation is linked to weather conditions would likely reap similar benefits. For example, the risk of wind droughts (periods of low wind, reducing wind generation) and hydrological droughts are likely uncorrelated globally, making a risk pool for those resources similarly attractive. To take our pooled approach one step further, a risk pool covering multiple, uncorrelated perils - such as wind, solar, and hydrologic droughts - could be established, boosting the cost effectiveness of the pool[42] and consequently augmenting the financial viability of weather-dependent renewable energies. Thermal generators are also impacted by extreme weather, including by droughts and heatwaves, as these can reduce access to cooling water, thereby inhibiting power generation

and leading to economic losses[49,50]. Inclusion of thermal generators in the risk pool would further grow the size thereof. That said, the sources' parallel vulnerability to drought[50] impact a country's risk contribution to the pool by increasing the magnitude of payouts during drought years. Expanding the number and variety of energy resources protected via the pool can also mitigate the risk of simultaneous failures in any one resource or changes in the spatial correlation of drought due to climate change. A final consideration may be including countries that rely on hydropower imports, for which a drought in a neighboring hydropower-dependent may have cascading financial impacts.

This work has several limitations. First is the level of basis risk and the number of countries in the pool (27% of countries reliant on hydropower), both of which can be attributed to the dearth of recent, coincident historical data. Though streamflow reanalysis datasets, e.g.,[31] or[51], can provide additional data, they offer poor geographic coverage and/or are limited to a few years, factors hindering the design of a global pool. In contrast, remotely sensed data provides greater spatial and temporal coverage. The dataset remains limited, however, meaning that it is unsurprising that indices have high levels of basis risk. Existing index insurance products similarly face challenges in managing basis risk, some of which can be mitigated by dynamically adjusting triggers and indices throughout the duration of an insurance contract[52,53]. This was the approach taken by CCRIF, whose self-described "undesirable" levels of basis risk were improved with continued iterations of its models[54]. In the context of this analysis, the indices developed serve as useful starting points, and additional financial and hydrometeorological data available within country is likely to lead to lower basis risk and more effective insurance contracts. Other emerging data sources may serve the same function. For example, the Surface Water and Ocean Topography Mission (SWOT) directly measures surface water, and early results suggest that it is able to capture even small variations in surface water extent and elevation[55], which could greatly improve index accuracy. Additionally, in a commercial setting, premiums would likely be re-evaluated either annually or within a multi-year contract. Such re-evaluation is increasingly important as global warming leads to changing likelihoods of hydrological drought around the world[56]. In this illustrative example, premiums are held constant. This work also omits consideration of political concerns. For example, countries risk domestic political backlash when premiums draw resources away from other budgetary priorities but fail to provide payouts. These factors have historically led to hesitation to purchase insurance and decisions to terminate contracts[11,40].

Future work might address these elements and incorporate costs beyond hydropower losses into insurance payouts by (1) considering the value of reliable electricity and (2) accounting for emissions damages from offsetting hydropower shortfalls with fossil fuel generation. With regards to the former, in regions and countries with substantial hydropower and limited alternative generating capacity, purchasing compensatory electricity to offset lost hydropower may be challenging. Droughts in such areas can thus contribute to rolling blackouts, which can have nation-wide economic consequences that go beyond lost hydropower revenues[57–59]. To that end, future analyses might attempt to model the combination of risk transfer tools (e.g., the risk pool) and risk reduction financing. Risk reduction projects, such as diversifying a hydropower-dominated generation portfolio by adding solar capacity to reduce vulnerability to drought, may pre-emptively help avoid or reduce blackouts before insurance payments are triggered, providing substantial, but difficult to quantify, advantages to countries that go beyond occasional insurance payouts. A second avenue for future work is expanding beyond direct financial losses to the hydropower sector to also quantify changes in emissions. Because hydropower is one of the most readily dispatchable renewable energy sources, when shortfalls occur they are often offset with fossil fuel

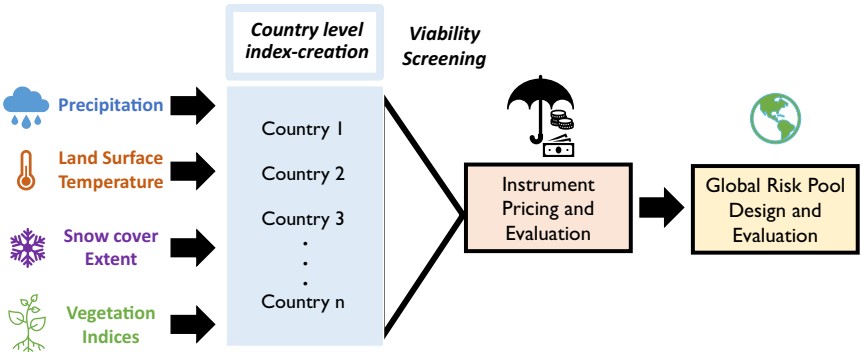

**Fig. 8 | Methodology flowchart.** Remotely sensed precipitation, land surface temperature, snow cover extent, and vegetation index data are used to construct national-level index insurance contracts for hydropower generation. The cost of insurance when individually held is compared to the cost when insurance contracts are pooled.

generation, leading to spikes in emissions from power plants[60]. Insurance payouts might be adjusted to include compensation for the indirect health costs of emissions, as well as any increased costs from fossil fuel generation. Insurers themselves might be incentivized to provide financing to install additional renewable energy capacity that can reduce a country's reliance on hydropower in such cases, in order to reduce their future payouts to the country. Such investments can prevent the vicious spiral noted by Turner et al.[61] - climate change-driven reductions in precipitation leading to reduced hydropower generation and increased construction of high-emission fossil fuel generators. A final direction for future work may explore the potential for simultaneous or sequential streamflow shortfalls and hydropower failures. Insurers calibrate the appropriate level of liquid reserves they hold based on the distribution of potential losses for any given year and this in turn influences premiums. Were the modeled distribution of losses to prove inaccurate (e.g., because of climate change), premiums collected could be insufficient to fully offset payouts. This challenge would be particularly acute if periods of simultaneous shortfalls across the pool, or sequential shortfalls that drew down reserves year-on-year, occurred more often than projected.

## Methods

This work involves the design and testing of individual, country-level index insurance contracts in terms of their ability to offset severe hydropower losses and their cost, followed by an examination of the potential for aggregating them into a global risk pool (Fig. 8). Index-based contracts are designed with the use of remotely sensed hydrometeorological data. The following sections describe the data used in this analysis (Section "Remotely sensed data"); the design of the indices (Section "Index insurance"); and the creation of the risk pool (Section "Global risk pool"). Analysis was undertaken using Python 3.7 using the geopandas (version 0.10.2), shapely (version 1.8.0), statsmodels (0.13.2), scikit-learn (version 1.0.2), and scipy (version 1.7.1) packages[62].

### Remotely sensed data

Data acquired via remote sensing can be used to overcome the paucity of on-the-ground observations in many countries, providing more spatially and temporally complete information on hydrometeorological conditions such as precipitation, snow cover extent, enhanced and normalized vegetation index, and LST. With respect to this analysis, the first three relate to water availability and hydropower supply, while temperature has a strong relationship with electricity demand[36,63] as well as with evapotranspiration and other water supply indicators (e.g., vegetation indices). Each of the remotely sensed datasets is globally available between 2000 and 2023 at various time steps ranging from sub-daily to monthly. These data are complemented by information on watershed boundaries and data on

global dam/reservoir locations and sizes which are used for aggregation of hydrometeorological inputs at different spatial scales during the index design process. Brief descriptions of the datasets are provided here, though readers are referred to the Supplementary Information, Section 2 and the original dataset documentation for more details on how the data are collected and processed.

The Integrated Multi-satellitE Retrievals for the Global Precipitation Measurement Mission (IMERG) algorithm integrates precipitation data collected from passive microwave estimates, microwave-calibrated infrared estimates, and precipitation gauges. This analysis utilizes the "Final" monthly $0.1 \times 0.1$ degree product, version 06[64], which is summed to a monthly time step from half-hourly observations.

Snow cover extent can provide a complementary measure of snowmelt magnitude and timing when paired with the IMERG measurements for precipitation levels. In many parts of the world, snowmelt accounts for a large fraction of streamflow[65], and its influence on hydropower can outweigh that of rainfall, making it important to isolate. We use monthly NDSI data at the $0.05 \times 0.05$ degree spatial scale (MOD10CM)[33], produced from daily MODIS observations. The NDSI, the normalized difference of green and shortwave infrared wavelengths, captures the relative percentage of high reflectance values typical of snow in visible bands, thereby producing an estimate of snow cover extent. It should be noted that NDSI can also capture variations in surface water extent, not just snow, meaning that NDSI may show variation even in areas that do not experience snowfall. Indices are thus manually screened to ensure NDSI is only used for countries with snowfall.

Vegetation indices reflect hydrologic conditions by describing vegetation density, a proxy for water availability. The vegetation index data used in this analysis (MOD13C2)[34] is produced from 16-day composites of the same MODIS observations as the NDSI data, aggregated to the monthly time step. The NDVI is similar to the NDSI in that it measures the difference between the red and near-infrared wavelengths. The EVI improves on the NDVI in areas with dense vegetation by accounting for soil and atmospheric influences, thereby "decoupling" the vegetation from the background canopy signal. In regions with dense vegetation, EVI has improved sensitivity that can more accurately characterize vegetation levels. On the other hand, EVI is very sensitive to topography and so may produce error-laden index values in areas with variable topographies. This analysis includes both NDVI and EVI in index creation to improve accuracy for countries with either of these features.

Finally, LST data, which measures the emissivity and "brightness" temperature of the surface in Kelvins is also derived from MODIS. The MODIS LST data (version 6) is collected daily at a 1 km × 1 km scale but this analysis uses the $0.05 \times 0.05$ degree, monthly aggregation (MOD11C3)[35]. These five datasets are aggregated at several temporal (i.e., seasonal and annual) and spatial scales. Temporal averaging is

undertaken at the seasonal (i.e., winter, spring, summer, fall) and annual timestep. Spatial averaging is based on hydrologic unit boundaries from the HydroBASINS dataset[66], which delineates sub-basins by the location where two river branches meet, given a minimum upstream area of 100 km². Units are categorized into 12 levels ranging from the continental to the local watershed levels, with sub-continental and regional level basin scales used for this analysis. Data are additionally averaged within country borders. See Supplementary Information, Section 2 for more details on spatial and temporal averaging, as well as for a map of the level 4 subbasins used for each country.

Additional data are obtained from the US Energy Information Administration for historical hydropower generation from 1980 to 2023[67] and from the International Renewable Energy Agency (IRENA) for generation capacity, from 2000 to 2023[68].

## Index insurance

Index insurance emerged as a tool to manage hydrometeorological financial risk as early as 1920 in India[69]. Unlike indemnity-based insurance, which requires quantification and on-site verification of losses, payouts from index insurance contracts are triggered when a pre-defined threshold of some established metric (the "index"), measured at a specific time and place, is crossed. Payouts may be fixed or vary according to the difference between the index threshold, or strike, and the observed value (i.e., the larger the gap, the larger the payout). This index-based structure typically reduces administrative costs and concerns over moral hazard because damages and payouts are not directly linked. Additionally, less reliance on lengthy and subjective judgements over losses can increase the speed of payouts[70]. Though index insurance has long been associated with weather-based perils impacting agriculture[69–75], its potential has been explored across many other economic sectors, including shipping and inland navigation[76–78]; urban water supply[79,80]; and energy[81–86]. Other perils that have been covered by index insurance include wildfires, earthquakes, hurricanes, cyclones, windstorms, and floods[11,44,87,88]. The most notable example of index insurance related to hydropower is the 2013 index insurance contract written to cover the hydropower-dependent, publicly-owned Uruguayan electricity utility, UTE, which provides up to 75% of the country's electricity via hydropower[2]. In this case, reinsurer Swiss Re collaborated with the World Bank to underwrite the contract, which triggered a payout to UTE when rainfall fell below an agreed upon level as measured in a prescribed manner[5]. The goal of this contract was to offset the increased costs of generating and/or importing electricity from alternative sources during periods of low hydropower generation.

This work aims to design index-based contracts for countries dependent on hydropower to offset the lost value of hydropower generation during years with shortfalls, but to do so while pooling the risks of multiple countries. The primary challenge when developing such a contract is identifying an index that is sufficiently correlated with the insured party's losses (i.e., with low basis risk), a process that can be limited by data availability[89]. Designing a strong index requires consistent and reliable data to identify triggers for payouts and to characterize the relationship between hydrometeorological conditions, hydropower generation and financial outcomes. A poorly designed index leads to high basis risk, meaning a low level of correlation between losses and payouts such that payouts are not be triggered when losses occur, or are triggered when losses are minimal. The former results in ineffective risk management and the latter in more expensive contracts.

The remotely sensed data described in Section "Remotely sensed data" are used to meet this requirement. Indices are designed using precipitation, LST, NDVI, EVI, and snow cover extent data as inputs to multivariate linear regressions predicting national average hydropower capacity factor, or the ratio of actual electricity generated to maximum potential generation. Use of the capacity factor accounts for changes in hydropower generation capacity over time, as countries build new hydropower infrastructure or, in some cases, remove it. Data are split into training (80%) and testing (20%) sets. Only countries whose indices have less than four statistically significant predictor variables (two-tailed $p$-values ≤0.05) and an $r^2 \geq 0.35$ are considered when correlating losses with payouts. The restriction on the number of inputs to each index is an attempt to limit overfitting, especially important considering the limited time series (21 years) on which indices are based, but also to increase transparency, an important consideration for buyers of index contracts[40]. The second condition is intended to ensure a commercially viable level of agreement between the index and reduced generation (and sales revenue). Existing index insurance instruments capture less than 30% of financial variation[37–39], suggesting that 35% would be acceptable to contract buyers in terms of basis risk. Index-based contracts are designed only for hydropower-reliant countries, defined here as relying on hydroelectricity for at least 25% of domestic generation, those which would face substantial financial consequences during periods of hydropower shortfalls. Additional statistical tests for normality, heteroscedasticity, and accuracy were conducted, and can be found in the Supplementary Information, Section 1.

A key component of index-based instruments is the strike, which is the threshold that when crossed leads to payouts. There is no single method to set a strike, but different probabilistic measures of risk can provide useful guidelines, including use of VaR, the value below which a given percent of outcomes (e.g., generation, net revenues) fall based on the historical distribution of losses. The strikes in this analysis are set to each country's 80% VaR of capacity factor, indicating the worst 20% of outcomes in terms of annual generation, with the assumption that less severe losses can be compensated for using other tools, such as tariff adjustments and reserve funds. The capacity factor is translated into generation by multiplying it against 2020 installed hydropower capacity (Eq. (2)).

Generation is converted into a financial value using the global average levelized cost of electricity (LCOE) for hydropower (Eq. (1)). The LCOE takes into account both operating and capital expenses throughout the lifetime of an electricity generator, and in 2022 was $44/MWh for hydropower[90]. Use of the average levelized cost provides a consistent evaluation of the value of hydropower generated globally and avoids reliance on national electricity tariffs that may be heavily subsidized and which are not consistently available. In practice, this value may be adjusted according to the LCOE of hydroelectricity in any given country. Historical capacity factor data is used to set a cap on payouts, using the 99.5% VaR:

$$v(\$) = (\text{avg}_i - \text{cf}_{i,y}) * \text{capacity}_i * 8760 * \text{LCOE} \tag{1}$$

$$\text{payout}_{i,y}(\$) = \begin{cases} 0 \leq v_{i,y} \leq \text{cap}_i, & \text{if } \text{cf}_{i,y} < k_i \\ 0, & \text{otherwise} \end{cases} \tag{2}$$

where $v$ is the predicted value for the financial shortfall country $i$ faces in year $y$; cf is the capacity factor as calculated by the index; and capacity (MW) is a country's hydropower capacity in 2020. Because there are no negative capacity factor values, the hydropower shortfall is calculated as the difference between the average capacity factor ($\text{avg}_i$) and the index-calculated capacity factor for a given year. This value is multiplied by maximum annual potential hydropower generation—the product of hydropower capacity (MW) and the number of hours in a year (8760)—and by the LCOE ($/MWh) to calculate ($v$), which is intended to offset the hydropower shortfall. The final payout is triggered by the strike, $k$ (set to the twentieth percentile of capacity factor), and capped using the 99.5% VaR for capacity factor ($).

The premiums for the index insurance contracts are the sum of a country's expected payouts and loading. The first component, the expected payouts, is intended to offset payouts for the insurer over the long-term. Loading then accounts for the costs an insurer accrues in developing and administering the contract, as well as a reasonable return for accepting the risk of losses (e.g., opportunity cost of holding large liquid reserves). The loading loosely represents the cost of an insurance contract to the buyer of the insurance because the buyer will, statistically, receive the value of the expected payouts from the insurer over the long-term, but there is no accounting for the true value of money held by an insurer.

Though historical data on losses can be used to calculate expected payouts, there are various ways to estimate loading and full premiums. For example, the CCRIF, which insures Caribbean nations against hurricane-related losses, fits limited historical data to loss exceedance curves, which are used to assess the probability of a loss greater than a given threshold. Information on the probability and magnitude of payouts informs premium calculation for a contract providing a given level of insurance coverage. The distribution of payouts can also be altered to more heavily weigh the tail (higher payouts) using a risk adjustment factor to generate a risk-adjusted distribution for which the expected payout represents the premium. The Wang Transform[41] has been used to make this risk adjustment and has been applied to insurance pricing in several sectors[16,82,83,91–93]:

$$\text{F}^{*}(x_i) = \Phi[\Phi^{-1}(F(x_i)) + \lambda] \qquad (3)$$

$$v_i = \mathbb{E}(\text{F}^{*}(x_i)) \qquad (4)$$

where $\Phi$ is a standard normal distribution, $x(\$)$ are the payouts for country $i$, and $\lambda$ is the risk adjustment factor used to approximate the market price of risk and generate a risk-adjusted distribution of payouts. The market price of risk, $\lambda$ is set to 0.25 for this analysis, a benchmark value used for pricing weather derivatives in the original work[41]. The expected value of this risk-transformed distribution is the premium ($v_i$), and the loading is the difference between the expected payout and the final premium.

When comparing the costs of risk transfer via insurance with risk retention via reserves, loading on the insurance contract (the annual cost) can be compared to the opportunity cost of holding reserves in some accessible (liquid) form. This analysis uses the average returns (6.03%) on 20-year US Treasury bonds between 1974 and 2024[94]. Fig 5 shows results using historical rates from a high return period (1974–1999, 8.5%) and a low return period (2000–2024, 3.3%). It highlights the influence of investment returns on the relative cost-effectiveness of purchasing insurance.

### Global risk pool

While index insurance can be effective for individual people or countries, bundling the uncorrelated (or negatively correlated) risks of several parties can reduce the costs of insuring all pool participants relative to a scenario in which each is independently insured or chooses to manage its risk via reserves (risk retention). As the degree of correlation between losses experienced by pool participants declines, the total reserves required by the pool to cover collective losses similarly falls far below what the parties in the pool would have to maintain independently[42].

Thus, one important factor in determining the effectiveness of a risk pool is the level at which losses are uncorrelated across participants. The degree to which the pool diversifies the risk of losses depends on this lack of correlation, and benefits from negative correlations[15]. Figure 7 shows correlations between annual hydropower capacity factors in hydropower-dependent countries. Although some countries experience correlations of up to 0.89, others

experience correlations as negative as −0.83, with a median correlation of 0.07. Though this work attempted to design indices for all countries depending on hydropower, only indices for 15 countries met the criteria of $r^2 \geq 0.35$, less than four indicator variables, and triggered payouts in at least 1 year. This subset of countries exhibits a range of bilateral correlations similar to the full set, with a range of 0.78 to −0.69 and a median correlation of 0.06. For individual countries, median bilateral correlations range between 0.15 and −0.16. The low, and often negative, median level of correlation suggests that a risk pool could be effective in reducing the aggregate reserves necessary to provide a defined level of risk management.

The pool comprises the individual country contracts, meaning that the annual pool payout is equivalent to the sum of annual payouts for all pool participants. The pooled premium is a single sum, priced by applying the Wang Transform to overall pool payouts. The difference between the premium for this pooled contract and the sum of individual premiums for all countries within the pool yields the savings from pooling countries' risk.

This pooled premium must then be disaggregated into individual premiums for pool participants. An important question then becomes how the savings associated with pooling should be distributed amongst pool members. In this work, savings are allocated according to each participant's average marginal contribution to the risk and diversification advantage that they contribute to the pool, in line with the literature on pricing individual components of a portfolio[95–100]. The marginal contribution is calculated as the Shapley value[101].

The Shapley value quantifies the average contribution of each pool participant to the cost of insurance, i.e., how the loading for the pooled contract changes based on a participant's addition to possible pool subsets. It incorporates changes in loading across all potential pool combinations, i.e., changes in loading were a country added to a pool of any size. Consideration of the average contribution allows the distribution of savings such that all members of the pool experience savings relative to the purchase of individual contracts or participation in a smaller pool (see Supplementary Information, Section 3). For example, a country that experiences payouts that are strongly correlated with other pool members will see a smaller relative reduction in its premium as a result of pooling compared to a country whose payouts are negatively correlated with other participants, even if their expected losses are equal. This is because the latter country provides greater diversification advantages to the pool.

Premiums within the pool are thus a function of the individual loadings calculated via the Wang Transform, and the Shapley value ($\varphi_i(N, v)$), which is the new loading value, accounting for the savings made possible by each pool participant:

$$\varphi_{i(N, v)} = \sum_{\substack{C \subseteq N \\ i \in C}} \frac{(|C| - 1)!(n - |C|)!}{n!} \{v_N - v_{N \setminus \{i\}}\} \qquad (5)$$

where $N$ is the full set of pool participants, $C$ is a permutation of that set, $i$ is a single participant, and $v_N$ is the loading associated with the set of pool participants as calculated via the Wang transform. Once the savings from the risk pool are allocated to individual participants, the cost of the pooling strategy can be evaluated in comparison to (1) the opportunity cost of holding liquid reserves sufficient to offset generation losses up to the maximum insurance payout (payout cap = 99.5% VaR) and (2) premiums that would be charged to pool members if they were to independently purchase insurance contracts.

### Reporting summary
Further information on research design is available in the Nature Portfolio Reporting Summary linked to this article.

## Data availability

Data on index insurance contract formulation and costs of risk management generated in this study have been deposited in a Zenodo repository (https://doi.org/10.5281/zenodo.17329375). Input data used in this analysis is publicly available and can be found on the Earthdata database: land surface temperature (10.5067/modis/mod11c3.061), precipitation (10.5067/gpm/imerg/3b-month/06), vegetation indices (10.5067/modis/mod13c2.061), and snow cover extent (10.5067/modis/mod10cm.061). Basin boundary data is taken from the HYBAS database and the Global Reservoir and Dam database provided coordinates and uses for dams (10.1890/100125). Additional data used in this study can be found in the Zenodo repository (10.5281/zenodo.17329375).

## Code availability

Code used for analysis is publicly available at https://github.com/rcuppari/Hydro_Risk_Pooling[62].

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

## Acknowledgements
This work was supported by the National Science Foundation's Dynamics of Coupled Natural and Human Systems (CNH) Program, award no. 2009726 (TMP and GWC) and the William R. Kenan Jr. Charitable Trust (GWC).

## Author contributions
All authors conceptualized the project and contributed to methodology and writing – reviewing and editing. Additionally, R.I.C. contributed to data curation, formal analysis/investigation, visualization, and writing – original draft. T.M.P. and G.W.C. contributed to funding acquisition.

## Competing interests
The authors declares no competing interests.
