## [Transparent Peer Review file · Nature Communications]

Global Risk Pooling Mitigates Financial Risk from Drought in Hydropower-Dependent Countries

Corresponding Author: Dr Rosa Cuppari

Version 0:

Reviewer comments:

Reviewer #1

(Remarks to the Author)

Dear Authors,

Thanks for this interesting submission which is methodologically robust and very policy relevant. I hope you will engage with relevant international bodies (e.g., International Hydropower Association) and financiers (e.g., World Bank) to share your framework and apply it in practice. I made a few minor suggestions in the attached file for your consideration.

Best regards,

Edoardo Borgomeo

(Remarks on code availability)

Library is well structured and readme file is ok.

Reviewer #2

(Remarks to the Author)

Dear Authors,

The study described in the manuscript titled ‘Using Remote Sensing to Develop a Global Risk Pool for Hydropower-dependent Countries’ develops hypothetical index insurance contracts to insure hydropower-dependent countries against financial risks associated to drought.

The study analyzes the potential benefits of these contracts for individual countries and within a global risk pool in comparison with self-insurance via a reserve fund. The results indicate that reasonably accurate contracts could be developed for 16 countries out of the 56 hydropower-dependent countries. For these 16 countries, pooling the index-insurance contracts could lead to relevant average savings (68%) relative to individual country risk management using reserves. Less significant comparative advantages are also estimated with individual index-insurance contracts, but the benefit is contingent on the level of average returns (interest rates) used to calculate the opportunity cost of reserves.

The topic of this study is certainly of interest for the disaster risk financing research and professional global community from an operational perspective. However, the paper misses a clear review of the state of the art of scientific research related to the specific topic and does not clearly highlight what the cutting-edge innovative scientific contribution of the study is, and why this can be relevant for a broad scientific audience. This is a major drawback.

The overall technical quality and readability of the manuscript is good, and results are presented and discussed in a clear way.

However, there are some major methodological issues to highlight:

- It is clear that for most hydropower-dependent countries considered (40 out of 56) the contract accuracy in predicting hydropower generation and revenues does not meet even the low requirements (i.e. model $R^2 > 0.35$ and one historical payout triggered) set in this study to assess the contract suitability for commercial implementation. This topic is barely discussed and analyzed.
- The requirements for index-insurance contract acceptability set in this study are low and, according to Authors, all contracts are “in the low end of acceptability of commercially available contracts”. To support the choice of the requirements, the

Authors cite selected paper from the index-insurance literature. However, these papers are all related to crop index-insurance and it is not very clear how the threshold was ultimately set. It seems that contracts which such a high uncertainty might not be easily acceptable for index-insurance schemes adopted at country level for a strategic and economically relevant sector like hydropower, especially for countries having access to relatively high-quality data. This uncertainty is also not taken into account in the comparison between index-insurance and self-insurance, while it might be a relevant factor to consider for the insurance pricing (as well for the willingness to purchase by countries). While the Authors correctly recognize that the quality of the proposed contract is a limitation of the study, a more in-depth analysis of basis risk would be essential to support the findings of this study and the broad suggested implications (i.e. that a pooled index-insurance product can be an effective tool for improving the financial stability of hydropower-dependent countries)

- In addition to the previous consideration, the description of the contract design is limited and R2 is the only performance indicator presented to compare multivariate models. This is a strong limitation, especially considering that multiple linear regression models are developed using a data limited time series (18 years), with risk of overfitting and violation of regression assumptions. A much more in-depth analysis of the model uncertainty would be important to support the potential effectiveness and commercial value of the proposed contracts.

- Remaining in statistical methods, the analysis of correlations between national level capacity factors is also unclear (Figure 4). In the text the Authors state that -0.84 is "a low correlation", while it is a high negative correlation. Then it seems that the average between negative and positive correlations is done, while the absolute values should be considered. Perhaps this is just a misunderstanding.

-It is also not very clear how the different remote sensing indicators were processed and aggregated spatially and temporarily for each of the country models. Some information is provided in Appendix A, but the details provided do not allow reproducing the indices and evaluating specific technical choices made in the contract technical design. This is a very critical step that can affect the overall quality of the index-insurance contract and should be presented and discussed in more detail.

Best regards,

(Remarks on code availability)

Reviewer #3

(Remarks to the Author)

The manuscript is well written, relevant, and valuable and provides a useful analysis for the audience.

I have two items for the authors to address:

1) My understanding is that there is an active weather derivatives market for hydropower, and note power derivatives in the early 2000s was part of the motivation for index insurance. In a quick search, I see that EEX European Hydro Power Go appears to be an active product. It may be that derivatives are a more appropriate mechanism for hydropower than insurance, but since the math stays the same, their paper is just as relevant. The authors should reference these markets, discuss their gaps and relevance, and connect their work to the derivatives market.

2) The key driving factor for price savings is how much of the total payouts in the portfolio are covered by the premiums from other reservoirs, as the remainder must be financed by the insurance company. The formulas and assumptions used in the paper are reasonable, but are not completely illustrative of this fundamental driver. They should highlight in their discussion how much of the payouts would come from provided through other reservoirs premiums, and how much would need to be financed by insurance companies comparing pooled and non-pooled setups based on historical burn, prior to assuming loading and distributional weighting.

(Remarks on code availability)

Version 1:

Reviewer comments:

Reviewer #1

(Remarks to the Author)

Dear authors,

I have reviewed your responses and revised manuscript, and I don't have any further comments. Thanks for Figure 2, which I think will help readers immediately understand the value of what's being proposed here.

It has been a great pleasure reading this work, and I wish you all the best with the dissemination. Thanks for your excellent submission.

Best regards,
Edoardo

(Remarks on code availability)

The code is available and the readme file well structured. Authors need to make sure they update the file names with the latest manuscript title.

Reviewer #2

(Remarks to the Author)

Dear Authors,

Thank you for addressing my comments in a careful manner. The manuscript quality and clarity have improved.

I have a couple of remaining points to clarify:

- On the cross-validation, the Authors mention they have used a leave one out cross-validation. However, in the Appendix A a 80%-20% splitting approach is mentioned. It is also unclear if the R2 presented in Figure 3 refers to cross-validated models or fitted models.

- The choice of a fixed season for temporal aggregation is somewhat questionable, especially in terms of rainfall distribution. The concept of spring, summer, fall, winter works mainly for mid-latitude countries, while it can be very different for tropical or equatorial countries. In addition, there are major geographic changes in seasonality due to a number of factors. Improving the temporal aggregation by defining the correct seasonality for each country might significantly improve the model performance.

Specific comments

Line 107: 27%

Line 181: 17 countries

Line 241: increases

(Remarks on code availability)

Reviewer #3

(Remarks to the Author)

The authors have addressed my points adequately for moving forward.

Although the requested analysis on risk pooling was not performed (as far as I could see), an adequate discussion was included.

I do believe that the manuscript would be of more value to the audience (including the re-insurers and risk pools now cited) if there was analysis on the reduction in risk costs if the reservoirs were pooled together verses priced individually.

I do not think that the time-dependent changes mentioned in the authors response are an important connection to this question.

(Remarks on code availability)

Response to Reviewers for: Using Remote Sensing to Develop a Global Risk Pool for Hydropower-Dependent Countries

We would like to express our appreciation to the reviewers and editors for their time and insights. The comments and questions have helped hone the message of the manuscript and clarify the original takeaways.

The responses below show **reviewer comments in blue font** and the authors' responses in regular black font. Reviewer comments refer to the line, section, and figure numbers of the original manuscript while references to changes made refer to the newly revised manuscript, without tracked changes. Any articles mentioned in the responses below are cited in a separate bibliography at the end of this document. Revised excerpts show added text in **red** and deleted text as **red, with a strikethrough**.

Reviewer #1

Dear Authors,

Thanks for this interesting submission which is methodologically robust and very policy relevant. I hope you will engage with relevant international bodies (e.g., International Hydropower Association) and financiers (e.g., World Bank) to share your framework and apply it in practice. I made a few minor suggestions in the attached file for your consideration.

Best regards,

Edoardo Borgomeo

Thank you Dr. Borgomeo – it is indeed our hope that this paper (at least the ideas underlying it) can be of practical use to international organizations, and we are grateful for this affirmation.

Reviewer #1 (Remarks on code availability):

Library is well structured and readme file is ok.

The paper 'Using remote sensing to develop a global risk pool for hydropower-dependent countries' presents an innovative framework to mitigate financial risks to hydropower production worldwide. The paper presents a significant advancement to the field, it's very well written, clear and methodologically robust. Having worked at the World Bank for a good part of the last decade, I can attest that this type of work has tremendous relevance for governments in low-income countries that rely on hydropower (or other renewable energy sources, as this framework can also be applied to wind production, for example). I think the paper should be accepted pending some minor revisions and clarifications. Below I provide some minor comments for the authors' consideration.

- Title. Titles are important and I think the current title does a disservice to the paper. Could the authors use a more direct title to convey the essence of the findings? For example, 'Global risk pooling helps mitigate climate-related financial risks for hydropower-dependent countries' or something like that.

We appreciate the point on the title and have modified it to:
Global Risk Pooling Can Mitigate Drought-Related Financial Risk for Hydropower-Dependent Countries

- Input data. I have gone through the data and methods section, and find the proposed data sets and approach appropriate given the global scope of the study. However, I was a bit surprised that the authors did not include any global streamflow dataset. For example, reanalysis datasets (e.g., Alfieri et al. 2020; Bosmans et al. 2022 for projections) or other hydrological datasets being developed for machine learning type of analysis (E.g., Chen et al. 2023). It would be great if the authors could explain briefly why they decided not to use streamflow data. I understand the focus is on remote sensing, but I think we agree that an index should be developed on best available evidence, irrespective of the data collection approach.

We did consider the use of reanalysis data but ultimately decided that it was important to use data that extended until at least 2020 to provide more recent overlap with other critical datasets used in this work. The reanalysis dataset constructed by Alfieri et al. (2020), for example, provides data from 1980, but ends in 2018. The Global Runoff Data Centre has additional useful hydrologic data, but it ends in 2019. Chen et al. (2023a) developed another notable dataset, but of the countries included, only 6 have data for streamflow stations between 2000 and 2020. Considering that the remotely sensed hydrometeorological data is consistently available from 2002 onwards, inclusion of any of these datasets would have reduced the length of our usable historical data.

In general, the satellite data used here has the advantage of being consistently and globally available for a longer coincident time period, thereby allowing for the development of indices that capture multiple conditions simultaneously. However, we recognize (and hope) that a future risk pool designed in partnership with participant countries would have access to more recent, higher resolution data. To reflect this reasoning in the text, we have added the following:

- Page 5, lines 122-124: First, country-level index contracts are designed using remotely sensed data which, while limited, provide extensive, coincident historical data with greater temporal and spatial coverage than alternative sources, including streamflow reanalysis datasets (e.g., Alfieri et al. (2020)).
- Page 15, lines 331-349: This work has several limitations. First is the level of basis risk and the number of countries in the pool (32% of hydropower-dependent countries), ~~associated with the index insurance contracts~~, both of which can be attributed to the dearth of recent, coincident historical data. Though streamflow reanalysis datasets, e.g., Alfieri et al. (2020) or Chen et al. (2023b), can provide additional data, they offer poor geographic coverage and/or are limited to a few years, factors hindering the design of a global pool. In contrast, remotely sensed data provides greater spatial and temporal coverage. The dataset remains limited, however, meaning that it is unsurprising that indices have high levels of basis risk. Existing index insurance products similarly face substantial challenges in managing basis risk, some of which can be mitigated by dynamically adjusting triggers and indices throughout the duration of an insurance contract (Palwishah et al., 2023;

Viergutz, 2024). This was the approach taken by CCRIF, whose self-described “undesirable” levels of basis risk were improved with continued iterations of its models (CCRIF, 2023). In the context of this analysis, ~~While~~ the indices developed serve as useful starting points; ~~the use~~ additional financial and hydrometeorological data available within country ~~is likely to may~~ lead to ~~stronger indices,~~ lower basis risk, and more effective insurance contracts. ~~Other emerging data sources may serve the same function.~~ For example, the Surface Water and Ocean Topography Mission (SWOT) directly measures surface water, and early results suggest that it is able to capture even small variations in surface water extent and elevation (Fu et al., 2024), which could greatly improve index accuracy.

- Assumption that drought risks are uncorrelated globally. While I would generally agree with the statement made by the authors that drought risks to hydropower are uncorrelated globally, there is also lots of research that examines joint occurrence of extremes globally, particularly focusing on food production (e.g., Gaupp et al. 2020; Hasegawa et al. 2022). These papers suggest that risks of simultaneous breadbasket failures cannot be ignored. I would therefore suggest that the authors reflect on this issue in their discussion and perhaps suggest refinements to their methodology that will improve the representation of potentially simultaneous hydropower failures.

Thank you for raising this issue, which will likely become even more pressing with a changing climate. Simultaneous hydropower shortfalls would present challenges for insurers, as correlated failures undermine the benefits of risk pooling. From an insurer’s perspective, simultaneous failures would increase the level of liquid capital needed to protect against a simultaneous failure. However, so long as there is not a strong correlation between hydropower shortfalls among all countries, pooling is still a valuable and cost-saving approach. In our subset of countries, bilateral correlations based on the historical data range between -0.66 and 0.83, but the median bilateral correlations for each country range between -0.17 and 0.22. The lower range of bilateral correlations suggests that even if national hydropower generation were slightly more correlated than the data show, pooling would still have value.

To address these comments, we have added the following text into the discussion

- page 14, lines 327-329: A final consideration may be including countries that are dependent on hydropower imports, on which a drought in a hydropower-dependent country may have cascading financial impacts.
- Page 15, lines 382-390: A second avenue for future work may explore the potential for simultaneous or sequential streamflow shortfalls and hydropower failures. Insurers calibrate the appropriate level of liquid reserves they hold based on the distribution of potential losses for any given year and this in turn influences premiums. Were the modeled distribution of losses to prove inaccurate (e.g., because of climate change), premiums collected could be insufficient to fully offset payouts. This challenge would be particularly acute if periods of simultaneous shortfalls across the pool, or sequential shortfalls that drew down reserves year-on-year, occurred more often than projected.

- Other aspects of the water-energy nexus. The approach proposed here would be very relevant also for thermal power generation, which in places like the US is very vulnerable to droughts (e.g., Van Vliet 2016). The authors might wish to mention similar additional applications.

The reviewer is correct in noting that this approach could be replicated across many sectors and locations. To that end, we have added text into the discussion page 14, lines 319-325:

Thermal generators are also impacted by extreme weather, including by droughts and heatwaves, as these can reduce access to cooling water and thereby inhibit power generation and lead to economic losses (Byers et al., 2020). Inclusion of thermal generators in the risk pool would further grow the size thereof. That said, the sources' parallel vulnerability to drought may impact a country's risk contribution to the pool by increasing the magnitude of payouts during drought years.

- Conceptual figure. The authors might wish to consider including a summary figure at the start of the paper that shows how financial risks to hydropower from drought can be managed, including through a global risk pool (such as Figure 1 in Mechler et al. 2014). This could expand what is currently Figure 5 of the methods.

Thank you for this suggestion. We have added a new figure on page 4 to illustrate different risk management approaches.

Figure 2. The risk of high probability, low impact events is typically managed using risk reduction or risk retention. Risk transfer via either individual insurance or risk pooling is best suited for low probability, high impact events. Dams represent hydropower-dependent countries.

- Figure 1. Did the authors search for any more recent data from the international energy agency on total generation from hydropower?

There is IEA data available through 2024 on hydropower generation, but it is only available upon payment. Our intent was to exclusively use open access data, recognizing that a pooling strategy, if implemented by an international finance institution in

partnership with individual countries, would rely on more data than is publicly available on a consistent basis. That said, we realize that the data had not been updated since we initiated the work, and the International Renewable Energy Agency has released electricity generation data through 2023 as of the time of this review (IRENA, 2024). The revisions to this analysis now include the updated data through 2023. Each index was refined and updated using the new data points, as can be seen in the revised Figure 3. Though two indices were dropped (Namibia and Sweden), we were able to develop indices for three additional countries (Norway, Panama, and Pakistan). All results have been updated accordingly, though the tables and figures are *not* highlighted in red. *Note:* to make the data consistent, we have opted to update all of the land surface temperature data to 0.05 degrees

Fig. 3 Performance of indices along with regression formula. Variables are averaged at the country level or, when ending with "_4", at the subbasin scale, using the subbasin with the greatest number of hydroelectric dams. Seasons reflect Northern Hemisphere months (e.g., winter indicates December-February).

- Link between r² and hydropower characteristics. Figure 2 and section 2.1 show that r² is >0.35 only for some countries. Is there any particular pattern behind the countries for which r² is >0.35? Are these countries where there are just a few, large hydropower facilities (e.g. Kariba in Zambia, Itaipu in Paraguay). It could be interesting for the authors

to assess whether there is any correlation between number/size of hydropower facilities and the performance of the indices.

We were also interested in identifying whether index creation was more successful in some groups of countries or not. We attempted to break countries into groups based on which variable most closely correlated with generation, or what season was most common, but there was no discernable pattern. We had not considered the number/size of hydropower facilities, which is an interesting question. The point proves true – 13 of the 17 countries for which indices were developed have fewer than three dams whose ‘Main’ or ‘Major’ purpose is to generate hydroelectricity (as categorized in the Global Reservoir and Dam Database database). One exception to this rule, Albania has a greater number of dams, but on fewer than three rivers. Of the pool participants, only Norway has both a large number of hydroelectric dams and a large geographic spread of these dams. Together, the number of hydroelectric dams and the number of rivers can be used in a logistic regression to predict the significant countries, with an accuracy of 80% and an area under the receiver-operating characteristic curve (ROC) of 66%. These metrics suggest the two factors are indicative of whether an index can be formulated for a country, though they are not conclusive; 14 additional hydropower-dependent countries have fewer than three hydroelectric dams though no viable index was created for them. Many thanks to the reviewer for raising this point.

To reflect on this point, we have added the following text (page 6, lines 159-169):

Of the countries considered, 176 have indices with r^2 values ≥ 0.35 (see Figure 2) and trigger at least one payout during the 21 year historical timeseries (2002-2023). This threshold with $r^2 \geq 0.35$ (see Figure 2), a level corresponding corresponds to the low end of commercially available index insurance contracts (Gong et al., 2023; Jensen et al., 2016; Tsay & Paulson, 2024), and trigger at least one payout during the 18-year historical time-series (2002-2020). Notably, 77% of the 18 countries in our pool have fewer than three dams whose ‘Main’ or ‘Major’ purpose is to generate hydroelectricity (as categorized in the Global Reservoir and Dam Database (GRaND) database). The remaining four participants (Norway, Albania, Croatia, and Georgia) have a greater number of dams, though only Norway has dams on more than 5 rivers, as designated by the GRaND database. This suggests that future efforts to form risk pools might initially focus on countries with concentrated hydroelectric facilities.

Reviewer #2

The study described in the manuscript titled ‘Using Remote Sensing to Develop a Global Risk Pool for Hydropower-dependent Countries’ develops hypothetical index insurance contracts to insure hydropower-dependent countries against financial risks associated to drought. The study analyzes the potential benefits of these contracts for individual countries and within a global risk pool in comparison with self-insurance via a reserve fund. The results indicate that reasonably accurate contracts could be developed for 16 countries out of the 56 hydropower-dependent countries. For these 16 countries, pooling the index-insurance contracts could lead to relevant average savings (68%) relative to individual country risk management using reserves. Less significant comparative advantages are also estimated with individual index-insurance contracts, but the benefit is contingent on the level of average returns (interest rates) used to

calculate the opportunity cost of reserves.

The topic of this study is certainly of interest for the disaster risk financing research and professional global community from an operational perspective. However, the paper misses a clear review of the state of the art of scientific research related to the specific topic and does not clearly highlight what the cutting-edge innovative scientific contribution of the study is, and why this can be relevant for a broad scientific audience. This is a major drawback.

The overall technical quality and readability of the manuscript is good, and results are presented and discussed in a clear way.

However, there are some major methodological issues to highlight:

- It is clear that for most hydropower-dependent countries considered (40 out of 56) the contract accuracy in predicting hydropower generation and revenues does not meet even the low requirements (i.e. model $R^2 > 0.35$ and one historical payout triggered) set in this study to assess the contract suitability for commercial implementation. This topic is barely discussed and analyzed.

The reviewer makes a good point, and we have added the following text to address these concerns (pages 14-15, lines 330-348). We would also like to note that use of two additional years of data enabled us to refine the pool, such that the total number of countries included is now 17. Greater availability of data, offered by host countries or collected over time, will continue to strengthen the indices and extend the effectiveness of the pool.

This work has several limitations. First is the level of basis risk and number of countries in the pool (32% of hydropower-dependent countries), associated with the index insurance contracts: both of which can be attributed to the dearth of recent, coincident historical data. Though streamflow reanalysis datasets, e.g., Alfieri et al. (2020) or Chen et al. (2023b), can provide additional data, they offer poor geographic coverage and/or are limited to a few years, factors hindering the design of a global pool. In contrast, remotely sensed data provides greater spatial and temporal coverage. The dataset remains limited, however, meaning that it is unsurprising that indices have high levels of basis risk. Existing index insurance products similarly face substantial challenges in managing basis risk, some of which can be mitigated by dynamically adjusting triggers and indices throughout the duration of an insurance contract (Palwishah et al., 2023; Viergutz, 2024). This was the approach taken by CCRIF, whose self-described “undesirable” levels of basis risk were improved with continued iterations of its models (CCRIF, 2023). In the context of this analysis, While the indices developed serve as useful starting points; additional financial and hydrometeorological data available within country is likely to may lead to stronger indices, lower basis risk, and more effective insurance contracts. Other emerging data sources may serve the same function. For example, the Surface Water and Ocean Topography Mission (SWOT) directly measures surface water, and early results suggest that it is able to capture even small variations in surface water extent and elevation (Fu et al., 2024), which could greatly improve index accuracy.

- The requirements for index-insurance contract acceptability set in this study are low and, according to Authors, all contracts are “in the low end of acceptability of commercially available contracts”. To support the choice of the requirements, the Authors cite selected paper from the index-insurance literature. However, these papers are all related to crop index-insurance and it is not very clear how the threshold was ultimately set.

The minimum r^2 requirements for the indices are based on available literature on implemented contracts. While the examples cited are indeed related to crop index insurance, that is because index insurance is most commonly used in agriculture, with major sub-national and international programs including from the US Department of Agriculture’s Risk Management Agency, the Global Index Insurance Facility, the Caribbean Catastrophe Risk Insurance Facility, and the African Risk Capacity. Part of the novelty of this paper is applying index insurance to hydropower at the national level, an application with very few historical examples. The primary example in the hydropower sector, is the insurance contract offered by the World Bank and Swiss Re for hydropower in Uruguay (Swiss Re, 2018; The World Bank, 2013). However, we have been unable to find detailed information on the index or the contract’s efficacy. A final point is that even with the levels of basis risk seen in our analysis, the pool is still able to provide payouts and reduce premiums (loading) for participant countries, a proof of concept that may justify countries’ interest in insurance and a pool. Put into practice, the insuring party would most likely have access to higher quality data from individual countries, which would reduce basis risk and further increase the value of the pool.

We would also like to clarify the meaning of the quote referenced by the reviewer: *Of the countries considered, 16 have indices with $r^2 \geq 0.35$ (see Figure 2), a level corresponding to the low end of commercially available index insurance contracts (Gong et al., 2023; Jensen et al., 2016; Tsay & Paulson, 2024), and trigger at least one payout during the 18-year historical time series (2002-2020).*

Our reference to the low end of commercially available index insurance contracts is linked to the minimum r^2 (0.35), not to each of the contracts designed. The contracts in this work have r^2 values ranging from 0.43 to 0.81. To clarify this, we have modified the sentence:

Of the countries considered, 176 have indices with r^2 values ≥ 0.35 (see Figure 2) and trigger at least one payout during the 21 year historical timeseries (2002-2023). This threshold with $r^2 \geq 0.35$ (see Figure 2), a level corresponding corresponds to the low end of commercially available index insurance contracts (Gong et al., 2023; Jensen et al., 2016; Tsay & Paulson, 2024), and trigger at least one payout during the 18-year historical time-series (2002-2020).

It seems that contracts which such a high uncertainty might not be easily acceptable for index-insurance schemes adopted at country level for a strategic and economically relevant sector like hydropower, especially for countries having access to relatively high-quality data. This uncertainty is also not taken into account in the comparison between index-insurance and self-insurance, while it might be a relevant factor to consider for

the insurance pricing (as well for the willingness to purchase by countries). While the Authors correctly recognize that the quality of the proposed contract is a limitation of the study, a more in-depth analysis of basis risk would be essential to support the findings of this study and the broad suggested implications (i.e. that a pooled index-insurance product can be an effective tool for improving the financial stability of hydropower-dependent countries)

The reviewer makes a fair point that basis risk is a major challenge for parametric and index insurance products. The pool we propose provides an illustrative case of the potential of pooling. Were countries to join a risk pool, it is natural that they would contribute higher resolution and quality data, thereby reducing the basis risk involved. Yet even with the notable levels of basis risk for some of the indices, the cost of pooled insurance is ~30% of the opportunity cost of holding reserves, when the strike is set to cover the same level of financial losses (seen in Figure 4 of the original manuscript, now Figure 5). The addition of higher quality data would only make index insurance more attractive, as the basis risk associated with the contracts falls. One long-running risk pool protecting against hurricane-related financial losses, the Caribbean Catastrophe Risk Insurance Facility (CCRIF), explicitly describes this evolution. A CCRIF report notes that the original models and indices had “undesirable” levels of basis risk (unquantified in the report), which were improved with each iteration of the insurance products (CCRIF, 2023).

The following text has been modified to address the reviewer’s comment:

- Pages 14-15, lines 330-348 This work has several limitations. First is the level of basis risk and number of countries in the pool (32% of hydropower-dependent countries), associated with the index insurance contracts, both of which can be attributed to the dearth of recent, coincident historical data. Though streamflow reanalysis datasets, e.g., Alfieri et al. (2020) or Chen et al. (2023b), can provide additional data, they offer poor geographic coverage and/or are limited to a few years, factors hindering the design of a global pool. In contrast, remotely sensed data provides greater spatial and temporal coverage. The dataset remains limited, however, meaning that it is unsurprising that indices have high levels of basis risk. Existing index insurance products similarly face substantial challenges in managing basis risk, some of which can be mitigated by dynamically adjusting triggers and indices throughout the duration of an insurance contract (Palwishah et al., 2023; Viergutz, 2024). This was the approach taken by CCRIF, whose self-described “undesirable” levels of basis risk were improved with continued iterations of its models (CCRIF, 2023). In the context of this analysis, While the indices developed serve as useful starting points; additional financial and hydrometeorological data available within country is likely to may lead to stronger indices, lower basis risk, and more effective insurance contracts. Other emerging data sources may serve the same function. For example, the Surface Water and Ocean Topography Mission (SWOT) directly measures surface water, and early results suggest that it is able to capture even small variations in surface water extent and elevation (Fu et al., 2024), which could greatly improve index accuracy.

- In addition to the previous consideration, the description of the contract design is limited and R2 is the only performance indicator presented to compare multivariate models. This is a strong limitation, especially considering that multiple linear regression models are developed using a data limited time series (18 years), with risk of overfitting and violation of regression assumptions. A much more in-depth analysis of the model uncertainty would be important to support the potential effectiveness and commercial value of the proposed contracts.

Although we omitted mention of this in the original text, we also conducted leave-one-out cross-validation, judging accuracy using mean absolute error (MAE). Only two countries had a $MAE \geq 0.10$ (Zimbabwe – 0.14, Uruguay – 0.10) with respect to the prediction for capacity factor, which ranges from 0 to 1.

We have added text to that effect on page 6, lines 171-174:

Additional leave-one-out cross-validation provides further confidence in these indices, as all indices have mean absolute errors of ≤ 0.14 , relative to the capacity factor range of 0-1. The Normalized Nash-Sutcliffe Efficiency for each index lies between 0.51 and 0.72, indicating adequate fit.

We have added additional statistical tests for normality in Appendix B. Most countries fail to reject the null hypothesis of normality in all tests, although some fail one out of three tests. While this adds uncertainty to the analysis, we do not believe it alters the conclusion that bundling the risk faced by hydropower-dependent countries can lead to reduced cost of risk management.

Appendix B Additional Statistical Tests

Indices were developed using a criteria of $r^2 \geq 0.35$ in the training and testing datasets as well as overall. Additional metrics were used to evaluate normality, heteroskedasticity, and accuracy, as seen in Table BI. Generation data for most countries shows evidence of normality, though some fail one out of three tests. While this adds uncertainty to the analysis, we do not believe it alters the conclusion that bundling the risk faced by hydropower-dependent countries can lead to reduced cost of risk management.

Table BI. Metrics evaluating hydropower capacity factor data normality and heteroscedasticity as well as index performance. The Anderson-Darling test is evaluated with respect to $p = 0.01$

	White Test	Breusch-Pagan Test	Shapiro-Wilk Test	D'Agostino-Pearson Test	Anderson-Darling Test	MAE	Normalized Nash-Sutcliffe Efficiency
Albania	0.69	0.36	0.44	0.59	normal	0.07	0.52417
Angola	0.20	0.12	0.02	0.31	normal	0.07	0.57
Chile	0.38	0.45	0.09	0.32	normal	0.05	0.737
Costa Rica	0.67	0.62	0.19	0.06	normal	0.03	0.66284
Croatia	0.19	0.40	0.28	0.43	normal	0.05	0.72864
Ecuador	0.40	0.81	0.51	0.40	normal	0.04	0.70617
Gabon	0.16	0.76	0.01	0.18	normal	0.08	0.62
Georgia	0.53	0.30	0.87	0.88	normal	0.02	0.67732
Latvia	0.71	0.80	0.73	0.55	normal	0.03	0.62356
Norway	0.64	0.23	0.37	0.27	normal	0.02	0.63348
Pakistan	0.46	0.63	0.04	0.26	normal	0.05	0.57454
Panama	0.26	0.54	0.04	0.21	normal	0.04	0.62603
Paraguay	0.31	0.42	0.05	0.18	normal	0.05	0.69738
Suriname	0.63	0.42	0.24	0.13	normal	0.06	0.58846
Tajikistan	0.28	0.06	0.01	0.17	normal	0.03	0.52542
Uruguay	0.69	0.32	0.22	0.17	normal	0.11	0.61503
Zambia	0.48	0.76	0.88	0.95	normal	0.04	0.52928
Zimbabwe	0.17	0.14	0.06	0.21	normal	0.14	0.52

- Remaining in statistical methods, the analysis of correlations between national level capacity factors is also unclear (Figure 4). In the text the Authors state that -0.84 is “a low correlation”, while it is a high negative correlation. Then it seems that the average between negative and positive correlations is done, while the absolute values should be considered. Perhaps this is just a misunderstanding.

Thank you for pointing out this misrepresentation – the average of the absolute values does not adequately convey the existence or lack of correlations. To be more clear, we have modified the text in two places:

- Pages 21-22, lines 592-606): Thus, one important factor in determining the effectiveness of a risk pool is the level at which losses are uncorrelated across participants. ~~The degree to which the pool diversifies the risk of losses depends on this lack of correlation, and benefits from negative correlations (Ciullo et al., 2023).~~, ~~and therefore the degree of diversification of risk achieved by the pool (Ciullo et al., 2023).~~ Figure 7 shows correlations between annual hydropower capacity factors in hydropower-dependent countries. Although some countries experience correlations of up to 0.89, others experience correlations as ~~negative low~~ as -0.83-4, with a median correlation of 0.75. ~~an average correlation of 0.06.~~ Though this work attempted to design indices for all hydropower-dependent countries, only indices for ~~18+6~~ countries met the criteria of $r^2 \geq 0.35$, with less than four indicator variables, and triggered payouts in at least one year. This subset of countries ~~exhibits a similar range of bilateral correlations compared to the full set has comparable levels of correlations, ranging from 0.89 to -0.72~~ with a range of 0.83 to -0.66 and, ~~with~~ a median correlation of 0.1. ~~of an average of -0.05.~~ For individual countries, median bilateral correlations range between 0.22 to -0.17 . The low, and often negative, median level of correlation suggests that a risk pool could be effective in reducing the aggregate reserves necessary to provide a defined level of risk management.
- It is also not very clear how the different remote sensing indicators were processed and aggregated spatially and temporarily for each of the country models. Some information is provided in Appendix A, but the details provided do not allow reproducing the indices and evaluating specific technical choices made in the contract technical design. This is a very critical step that can affect the overall quality of the index-insurance contract and should be presented and discussed in more detail.

Thanks to the reviewer for pointing this out. An explicit listing of the steps is provided below and has now been added to the appendix with a parenthetical comment in the main body of the text pointing the reader toward the appendix (page 26, lines 701-716).

In summary, the indices were designed using the following steps:

1. Calculate the annual capacity factor for each hydropower-dependent country
2. Extract the average precipitation, land surface temperature, snow cover extent, and vegetation indices across two spatially bounded areas at the temporal level of the data (monthly). Data was not additionally filtered after download.
 - a. Within country (political borders)
 - b. Within the boundaries of the Level 4 subbasin in which the majority of a country's dams reside
3. Temporally aggregate the spatially aggregated data in two ways:
 - a. Aggregate at the seasonal level (i.e., winter, spring, summer, fall)

- b. Take the annual average of the spatially aggregated data
4. Evaluate the accuracy of a linear regression using any combination of variables. A training/test set split of 80/20 was used. This process was done automatically using a Python script that saved the combination meeting the test criteria of $\geq 0.35 r^2$ and less than four input variables.
5. Manually verify the accuracy of each index

Other aggregations were tested. For example, the maximum of the hydrometeorological variables as opposed to the average and a seasonal aggregation (fall/winter/spring/summer), but these aggregations did not result in more accurate indices, and so they were omitted in the final analysis.

To clarify the process used to aggregate each indicator, we have also added the following:

- Pages 19-20, lines 534-549, description of payout equation (note, the equation order has been reversed):

~~The~~ Historical capacity factor dataset is used to set a cap on payouts, using the 99.5% VaR:

$$v_{i,y}(\$) = (avg_i - cf_{i,y}) * capacity_i * 8760 * LCOE$$

$$payout_{i,y} = \begin{cases} 0 & \text{if } cf_{i,y} < k \\ \leq v_{i,y} \leq cap_i & \text{otherwise} \end{cases}$$

where v is the predicted value for the financial shortfall country i faces in year y ; cf is the capacity factor as calculated by the index; and $capacity$ (MW) is a country's hydropower capacity in 2020. ~~dollar value of the insurance contract's payout; cap (\$)~~ is the maximum payout for country i , set using the 99.5% VaR for capacity factor; ~~cf is the capacity factor as calculated by the index in year y ; k is the strike (set to the 20th% of capacity factor); and $capacity$ (MW) is a country's hydropower capacity in 2020.~~ Because there are no negative capacity factor values, the hydropower shortfall is calculated as the difference between the average capacity factor (avg_i) and the index-calculated capacity factor for a given year. This value is multiplied by maximum annual potential hydropower generation -- the product of hydropower capacity (MW) and the number of hours in a year (8760) -- and by the LCOE (\$/MWh) to calculate the insurance payment v , which is intended to offset the hydropower shortfall. The final payout is triggered by the strike, k (set to the 20th% of capacity factor) and capped using the 99.5% VaR for capacity factor cap (\$).

- Pages 17-18, lines 449-458, in reference to the snow cover extent, vegetation index, land surface temperature, and precipitation datasets): These five datasets are aggregated at several temporal (~~i.e., seasonal and annual~~) and spatial scales. Temporal averaging is undertaken at the seasonal (i.e., winter, spring, summer, fall) and annual timestep. Spatial averaging is based on hydrologic unit boundaries from the HydroBASINS dataset (Lehner & Grill,

2013), which delineates subbasins by the location where two river branches meet, given a minimum upstream area of 100 km². Units are categorized into 12 levels ranging from the continental to the local watershed levels, with sub-continental and regional level basin scales used for this analysis. Data is additionally averaged within country borders. See Appendix A for more details on spatial and temporal averaging, as well as for a map of the level 4 subbasins used for each country.

- Pages 25-26, lines 682-700): The above hydrometeorological data is aggregated temporally and spatially to provide possible inputs to an index. Temporally, data is averaged over the monthly and seasonal timescales. Use of the maximum and minimum values for each variable were also tested, but use of these variables provided no improvement on the indices. The data is then spatially aggregated in two ways: using country borders and subbasin boundaries. The HydroBASINS dataset developed by (Lehner & Grill, 2013) identifies different sizes of subbasins using the location where two river branches meet, given a minimum upstream area of 100 km². Subbasins Units are categorized into 12 levels according to the Pfafstetter system, ranging from the continental to local watershed levels. Level 4, which represents regional level basins, is used in this analysis. Hydrometeorological variables are averaged at each of these hydrologic units. The single level 4 basin within each country with that has the largest concentration of hydroelectric dams is identified in order to spatially aggregate environmental condition inputs. The Global Reservoir and Dam Database (GRanD) is used for this purpose (Lehner et al., 2011). It includes records on nearly 7,000 reservoirs and their dams, as well as information on their purposes. Only dams whose primary purpose is hydropower generation are included in the analysis. Once the level 4 subbasin of interest is identified, available, pixel-level hydrometeorological data (i.e., temperature, precipitation, snow cover, and vegetation indices) is averaged across the entire subbasin. The variables are averaged within country borders as well, providing a single average hydrometeorological value at two spatial scales and two temporal scales. In summary, two spatial scales are used as inputs to the indices: average environmental conditions at the national level and at the subbasin level.

Figure A1. Map with level 4 subbasins used to spatially aggregate remotely sensed data inputs highlighted in purple.

Reviewer #3

The manuscript is well written, relevant, and valuable and provides a useful analysis for the audience.

I have two items for the authors to address:

- 1) My understanding is that there is an active weather derivatives market for hydropower, and note power derivatives in the early 2000s was part of the motivation for index insurance. In a quick search, I see that EEX European Hydro Power Go appears to be an active product. It may be that derivatives are a more appropriate mechanism for hydropower than insurance, but since the math stays the same, their paper is just as relevant. The authors should reference these markets, discuss their gaps and relevance, and connect their work to the derivatives market.

As the reviewer notes, the math underlying derivatives and insurance is, in practice, identical. An important distinction between the two, however, might be that weather derivatives are publicly traded, while index-based insurance contracts are sold on an individual basis either as bespoke contracts or through individual agencies (e.g., the US Department of Agriculture's Risk Management Agency).

Thank you for bringing the EEX Hydro Power GO product to our attention. The GO (Guarantees of Origin) products are used to attribute the generation source used for a given unit of electricity meeting demand for buyers seeking to source a certain percentage of electricity from renewable energy sources. Their stated purpose is to track energy production (eex group, 2024), and so they appear to fulfill a different purpose than insurance or similar derivatives.

To address some of these points, we have added the following to the Introduction (**page 4, lines 92-95**):

The pre-set trigger and payment structure can also reduce administrative costs for the insurer and therefore the buyer. **Contracts using weather-based indices can be compared to weather-based derivatives** (Hertzler, 2004; The World Bank, 2023; Zeng, 2000), **which emerged in the 2000s to mitigate risk for power utilities and generators experiencing fluctuations in electricity use (and revenues) based on daily temperatures.**

2) The key driving factor for price savings is how much of the total payouts in the portfolio are covered by the premiums from other reservoirs, as the remainder must be financed by the insurance company. The formulas and assumptions used in the paper are reasonable, but are not completely illustrative of this fundamental driver. They should highlight in their discussion how much of the payouts would come from provided through other reservoirs premiums, and how much would need to be financed by insurance companies comparing pooled and non-pooled setups based on historical burn, prior to assuming loading and distributional weighting. As the reviewer surmised, this analysis assumes sufficient initial capital to provide payouts for countries in year 1, when premiums paid may be insufficient to cover payouts. In addition, we do not model time-dependent changes in premia, which can be triggered by the previous year's payouts when/if the total capital available to insurers falls (Heinrich et al., 2022). We believe the analysis remains reasonable and significant regardless because a risk pool of this magnitude (the national scale) would most likely be implemented by large international (re)insurers, perhaps in conjunction with international finance institutions like the World Bank. This category of (re)insurer has large, extant capital supplies and a well-diversified portfolio, able to absorb the risk of an additional, uncorrelated loss (i.e., drought for hydropower). The 2018 index insurance contract deployed to manage the impacts of droughts on Uruguay's hydropower followed this model, as it was developed by Swiss Re and the World Bank (Swiss Re, 2018). Swiss Re similarly worked with the Government of Mexico and the World Bank to insure natural perils via The Fund for Natural Disasters of Mexico (FONDEN) (Difiore & Drui, 2020; World Bank, 2020). We have added the following on pages 13-14, lines 291-303:

Countries are heeding these recommendations, joining regional risk pools like CCRIF and PCRIF or purchasing index-based instruments to manage the risk of natural hazards such as droughts, earthquakes, floods, tsunamis, and windstorms (Cooney et al., 2022; InsuResilience, 2021). **Partnerships between a large international finance institution (e.g., the World Bank) and large (re)insurers (e.g., Swiss Re) have enabled the development of these tools, taking advantage of the financial stability of the former in combination along with the access to capital and well-diversified portfolio of the latter. This type of partnership has been applied to insure Uruguay's hydropower sector against the risk of drought (Swiss Re, 2018), as well as to insure against multiple natural hazards in Mexico, via The Fund for Natural Disasters of Mexico (FONDEN)**

(Difiore & Drui, 2020; World Bank, 2020). At a broader level, national- and regional-level risk pools are exploring joint reinsurance facilities to further diversify their risk (Evans, 2023) reducing the levels of needed reserves and therefore lowering costs of risk management to pool participants.

Bibliography

Alfieri, L., Lorini, V., Hirpa, F. A., Harrigan, S., Zsoter, E., Prudhomme, C., & Salamon, P. (2020). A global streamflow reanalysis for 1980-2018. *Journal of Hydrology*, *X*, 6, 100049. <https://doi.org/10.1016/j.hydroa.2019.100049>

Byers, E. A., Coxon, G., Freer, J., & Hall, J. W. (2020). Drought and climate change impacts on cooling water shortages and electricity prices in Great Britain. *Nature Communications*, *11*(1), 2239. <https://doi.org/10.1038/s41467-020-16012-2>

CCRIF. (2023, February). *The Evolution of CCRIF's Parametric Insurance Models*. <https://www.ccrif.org/sites/default/files/DRF-Course-2023/CCRIFModel-Evolution-February2023.pdf>

Chen, X., Jiang, L., Luo, Y., & Liu, J. (2023a). A global streamflow indices time series dataset for large-sample hydrological analyses on streamflow regime. *Science Data Bank*. <https://doi.org/10.57760/sciencedb.07227>

Chen, X., Jiang, L., Luo, Y., & Liu, J. (2023b). *A global streamflow indices time series dataset for large-sample hydrological analyses on streamflow regime (until 2021)*. <https://doi.org/10.5194/essd-2023-49>

Ciullo, A., Strobl, E., Meiler, S., Martius, O., & Bresch, D. N. (2023). Increasing countries' financial resilience through global catastrophe risk pooling. *Nature Communications*, *14*(1), 922. <https://doi.org/10.1038/s41467-023-36539-4>

Cooney, N., Rajput, S., Hagemann, S., Villalobos, J.-A., & Bank, T. (2022). *InsuResilience* (Case Study No. 10). InsuResilience Global Partnership.

Difiore, P., & Drui, C. (2020). *Parametric Insurance: Beneficial By Nature*. Neuberger Berman.

eex group. (2024). *FAQ on Guarantees of Origin Futures*. eex group.

Evans, S. (2023, February). *Ping An launches parametric ocean carbon sink index insurance*. Artemis.Bm. <https://www.artemis.bm/news/ping-an-launches-parametric-ocean-carbon-sink-index-insurance/>

Fu, L., Pavelsky, T., Cretaux, J., Morrow, R., Farrar, J. T., Vaze, P., Sengenès, P., Vinogradova-Shiffer, N., Sylvestre-Baron, A., Picot, N., & Dibarboure, G. (2024). The surface water and

ocean topography mission: A breakthrough in radar remote sensing of the ocean and land surface water. *Geophysical Research Letters*, 51(4). <https://doi.org/10.1029/2023GL107652>

Gong, X., Hennessy, D. A., & Feng, H. (2023). Systemic risk, relative subsidy rates, and area yield insurance choice. *American Journal of Agricultural Economics*, 105(3), 888–913. <https://doi.org/10.1111/ajae.12342>

Heinrich, T., Sabuco, J., & Farmer, J. D. (2022). A simulation of the insurance industry: the problem of risk model homogeneity. *Journal of Economic Interaction and Coordination*, 17(2), 535–576. <https://doi.org/10.1007/s11403-021-00319-4>

Hertzler, G. (2004). Weather Derivatives and Yield Index Insurance As Exotic Options. *PSI Structural Genomics Knowledgebase*. <https://doi.org/10.22004/ag.econ.58705>

InsuResilience. (2021). *CCRIF SPC & World Bank Multi-Donor Trust Fund – InsuResilience Annual Report* (L. Gille, L. S. Kulick, D. Torkutsah, & A. Zwick, Eds.). InsuResilience Global Partnership.

IRENA. (2024). *Renewable energy statistics 2024*. International Renewable Energy Agency.

Jensen, N. D., Barrett, C. B., & Mude, A. G. (2016). Index Insurance Quality and Basis Risk: Evidence from Northern Kenya. *American Journal of Agricultural Economics*, 98(5), 1450–1469. <https://doi.org/10.1093/ajae/aaw046>

Lehner, B., & Grill, G. (2013). Global river hydrography and network routing: baseline data and new approaches to study the world's large river systems. *Hydrological Processes*, 27(15), 2171–2186. <https://doi.org/10.1002/hyp.9740>

Lehner, B., Liermann, C. R., Revenga, C., Vörösmarty, C., Fekete, B., Crouzet, P., Döll, P., Endejan, M., Frenken, K., Magome, J., Nilsson, C., Robertson, J. C., Rödel, R., Sindorf, N., & Wisser, D. (2011). High-resolution mapping of the world's reservoirs and dams for sustainable river-flow management. *Frontiers in Ecology and the Environment*, 9(9), 494–502. <https://doi.org/10.1890/100125>

Palwishah, R. I., Mazviona, B., & Sølvsten, S. (2023, December). *Addressing Basis Risk in Parametric Insurance for Disaster Resilience*. WTW. <https://www.wtwco.com/en-us/insights/2023/12/enhancing-disaster-resilience-addressing-basis-risk-in-parametric-insurance>

Swiss Re. (2018). *Closing the gap Drought threat to Uruguay's reliance on hydropower*. Swiss Re.

The World Bank. (2013, December 19). *The World Bank Partners with Uruguay to Execute Largest Public Weather and Oil Price Insurance Transaction*. The World Bank. <https://www.worldbank.org/en/news/press-release/2013/12/19/world-bank-uruguay-public-weather-oil-price-insurance-transaction>

The World Bank. (2023). *Index-Based Weather Derivative*. The World Bank.

Tsay, J.-H., & Paulson, N. D. (2024). Quantifying basis risk associated with supplemental area-based crop insurance. *Accounting and Finance Research*. <https://doi.org/10.1108/AFR-10-2023-0145>

Viergutz, A. (2024). *Basis risk in parametric insurance: challenges and mitigation strategies*. PwC.

World Bank. (2020). *World Bank Catastrophe Bond Provides Financial Protection to Mexico for Earthquakes and Named Storms*. <https://www.worldbank.org/en/news/press-release/2020/03/09/world-bank-catastrophe-bond-provides-financial-protection-to-mexico-for-earthquakes-and-named-storms>

Zeng, L. (2000). Weather derivatives and weather insurance: concept, application, and analysis. *Journals.Ametsoc.Org*.

Response to Reviewers for: Global Risk Pooling Can Mitigate Drought-Related Financial Risk for Hydropower-Dependent Countries

Dear reviewers,

Thank you for reviewing our paper once more. We are glad we were able to address most of your comments and hope you will find the additional revisions satisfactory. Your comments continue to strengthen the manuscript and its future utility.

The responses below show **reviewer comments in blue font** and the authors' responses in regular black font. Reviewer comments refer to the line, section, and figure numbers of the original manuscript while references to changes made refer to the newly revised manuscript, without tracked changes. Any articles mentioned in the responses below are cited in a separate bibliography at the end of this document.

Reviewer #1 (Remarks to the Author):

Dear authors,

I have reviewed your responses and revised manuscript, and I don't have any further comments. Thanks for Figure 2, which I think will help readers immediately understand the value of what's being proposed here.

It has been a great pleasure reading this work, and I wish you all the best with the dissemination. Thanks for your excellent submission.

Best regards,
Edoardo

Thank you Edoardo! We appreciate your comments. The figure was an excellent idea and we expect it will be useful for many a presentation in addition to this paper.

Reviewer #1 (Remarks on code availability):

The code is available and the readme file well structured. Authors need to make sure they update the file names with the latest manuscript title.

The title on the GitHub has been updated – thank you for catching that!

Reviewer #2 (Remarks to the Author):

Dear Authors,

Thank you for addressing my comments in a careful manner. The manuscript quality and clarity have improved.

I have a couple of remaining points to clarify:

- On the cross-validation, the Authors mention they have used a leave one out cross-validation. However, in the Appendix A a 80%-20% splitting approach is mentioned. It is also unclear if the R2 presented in Figure 3 refers to cross-validated models or fitted models.

Thank you for the suggestions to clarify our process. We first fit a model using the 80%-20% training-testing approach. Once the model was fit, we then used LOO cross-validation to ensure that model accuracy remained acceptable across any 80/20 split. The r^2 refers to the fitted model while the MAE score is the average across the LOO cross-validated models. Section 2.1 has been edited to clarify that the minimum r^2 threshold refers to the fitted model, while the leave-one-out cross-validation refers to a range of training and test set combinations.

The minimum r^2 threshold for the fitted model limits basis risk, which represents the ability of an index to trigger payouts when losses occur and at an appropriate amount. Additional leave-one-out cross-validation provides further confidence in these indices, as all indices have mean absolute errors (MAE) of ≤ 0.14 across a range of training and testing set combinations, relative to the capacity factor range of 0-1. The Normalized Nash-Sutcliffe Efficiency for each index lies between 0.52 and 0.74, indicating adequate fit.

- The choice of a fixed season for temporal aggregation is somewhat questionable, especially in terms of rainfall distribution. The concept of spring, summer, fall, winter works mainly for mid-latitude countries, while it can be very different for tropical or equatorial countries. In addition, there are major geographic changes in seasonality due to a number of factors. Improving the temporal aggregation by defining the correct seasonality for each country might significantly improve the model performance.

We also struggled with how to temporally group data such that we could capture seasonal effects without creating an excessive number of potential inputs, especially because we expected that annual values would be more reflective of hydropower capacity factor than seasonal alone. We settled on the three-month chunks (seasons) noting that while the label “winter” for the months of December-February refers to the Northern hemisphere, the three-month chunk could remain relevant even where it is in reality summer, and where two seasons mattered (e.g., fall into winter), those could be captured as two input variables to the index.

However, as the reviewer notes, the three-month chunks do not necessarily reflect hydrometeorological patterns. To that end, we re-evaluated the seasonal groupings for Angola, Costa Rica, Ecuador, Panama, Suriname, Zambia, and Zimbabwe – the countries in our pool with dry/rainy seasons that do not align with the four-season aggregations previously used in the analysis. None of the indices improved performance with use of a dry/rainy aggregation, neither in terms of r^2 nor in terms of statistical significance. Some worsened to a degree. We see two intuitive reasons for this. First, in many cases, the dry versus wet season split is equivalent to the average of a winter + spring versus summer + fall (e.g., Costa Rica’s December – April dry season and May – November rainy season). In these cases, a country’s index might contain inputs aggregated by two seasons. Second, in countries with substantial reservoir storage, annual values for precipitation may capture hydropower potential adequately, since the timing of hydropower generation is not dependent on the timing of rainfall.

Specific comments

Line 107: 27% corrected

Line 181: 17 countries corrected

Line 241: increases corrected, as “the costs of risk management...increase”

Reviewer #3 (Remarks to the Author):

The authors have addressed my points adequately for moving forward.

Although the requested analysis on risk pooling was not performed (as far as I could see), an adequate discussion was included.

I do believe that the manuscript would be of more value to the audience (including the reinsurers and risk pools now cited) if there was analysis on the reduction in risk costs if the reservoirs were pooled together versus priced individually.

I do not think that the time-dependent changes mentioned in the authors response are an important connection to this question.

Thank you, Reviewer #3, for your feedback during both rounds of revisions. Your suggestions improved the paper and gave us inspiration for future directions. We have made edits to Section 2.2. (below) in order to more clearly address how the magnitude of necessary reserves changes when they are held individually versus pooled, as we agree this is an important point driving the remainder of our findings.

The risks of the individual countries, and individual insurance contracts, can be pooled, with premiums modified to reflect the new, aggregate risk of the pool. The impact of pooling insurance contracts mirrors the reductions in costs when reserves are pooled instead of individually held.

This is because as the number of pool participants grows, the coefficient of variation associated with the aggregate payout falls, even as the overall, annual expected payouts increase. This can be attributed to the relatively low levels of correlation between each country's hydropower generation and drives the difference between the reserves necessary to compensate for the 99.5% VaR of the pool and the sum of the reserves countries would need to individually hold in order to independently manage their risk. Figure 5 shows the reduction in reserves achievable with multiple countries in the pool, whose cumulative reserves across all potential participants, if held individually, would amount to \$3.9 billion. Instead, when reserves are jointly held within a pool, the necessary amount falls by approximately 33% (\$1.3 billion) to \$2.6 billion. This corresponds to smaller reserves that an insurer would need to maintain as well, and in a competitive market lower costs for an insurer should lead to lower loading for purchasers (Bollmann & Wang, 2019).

Fig. 5: Pooling reserves reduces the levels needed to meet the 99.5% VaR of participating countries when compared to these countries individually holding reserves. Increasing the number of participants in a pool leads to greater savings. Values by pool size are calculated as the average savings across different combinations of pool participants.

Bibliography

Bollmann, A., & Wang, S. S. (2019). *International Catastrophe Pooling for Extreme Weather: An Integrated Actuarial, Economic and Underwriting Perspective*. Society of Actuaries.

Review of NCOMMS-24-73316

The paper 'Using remote sensing to develop a global risk pool for hydropower-dependent countries' presents an innovative framework to mitigate financial risks to hydropower production worldwide. The paper presents a significant advancement to the field, it's very well written, clear and methodologically robust. Having worked at the World Bank for a good part of the last decade, I can attest that this type of work has tremendous relevance for governments in low-income countries that rely on hydropower (or other renewable energy sources, as this framework can also be applied to wind production, for example). I think the paper should be accepted pending some minor revisions and clarifications. Below I provide some minor comments for the authors' consideration.

- **Title.** Titles are important and I think the current title does a disservice to the paper. Could the authors use a more direct title to convey the essence of the findings? For example, 'Global risk pooling helps mitigate climate-related financial risks for hydropower-dependent countries' or something like that.
- **Input data.** I have gone through the data and methods section, and find the proposed data sets and approach appropriate given the global scope of the study. However, I was a bit surprised that the authors did not include any global streamflow dataset. For example, reanalysis datasets (e.g., Alfieri et al. 2020; Bosmans et al. 2022 for projections) or other hydrological datasets being developed for machine learning type of analysis (E.g., Chen et al. 2023). It would be great if the authors could explain briefly why they decided not to use streamflow data. I understand the focus is on remote sensing, but I think we agree that an index should be developed on best available evidence, irrespective of the data collection approach.
- **Assumption that drought risks are uncorrelated globally.** While I would generally agree with the statement made by the authors that drought risks to hydropower are uncorrelated globally, there is also lots of research that examines joint occurrence of extremes globally, particularly focusing on food production (e.g., Gaupp et al. 2020; Hasegawa et al. 2022). These papers suggest that risks of simultaneous breadbasket failures cannot be ignored. I would therefore suggest that the authors reflect on this issue in their discussion and perhaps suggest refinements to their methodology that will improve the representation of potentially simultaneous hydropower failures.
- **Other aspects of the water-energy nexus.** The approach proposed here would be very relevant also for thermal power generation, which in places like the US is very vulnerable to droughts (e.g., Van Vliet 2016). The authors might wish to mention similar additional applications.
- **Conceptual figure.** The authors might wish to consider including a summary figure at the start of the paper that shows how financial risks to hydropower from drought can be managed, including through a global risk pool (such as Figure 1 in Mechler et al. 2014). This could expand what is currently Figure 5 of the methods.
- **Figure 1.** Did the authors search for any more recent data from the international energy agency on total generation from hydropower?
- **Link between r^2 and hydropower characteristics.** Figure 2 and section 2.1 show that r^2 is >0.35 only for some countries. Is there any particular pattern behind the countries for which r^2 is >0.35 ? Are these countries where there are just a few, large hydropower facilities (e.g. Kariba in Zambia, Itaipu in Paraguay). It could be interesting for the authors to assess whether there is any correlation between number/size of hydropower facilities and the performance of the indices.

References

- Alfieri, L., Lorini, V., Hirpa, F. A., Harrigan, S., Zsoter, E., Prudhomme, C., & Salamon, P. (2020). A global streamflow reanalysis for 1980–2018. *Journal of Hydrology X*, 6, 100049.
- Bosmans, J., Wanders, N., Bierkens, M. F., Huijbregts, M. A., Schipper, A. M., & Barbarossa, V. (2022). FutureStreams, a global dataset of future streamflow and water temperature. *Scientific Data*, 9(1), 307.
- Gaupp, F., Hall, J., Hochrainer-Stigler, S., & Dadson, S. (2020). Changing risks of simultaneous global breadbasket failure. *Nature Climate Change*, 10(1), 54-57.
- Hasegawa, T., Wakatsuki, H., & Nelson, G. C. (2022). Evidence for and projection of multi-breadbasket failure caused by climate change. *Current Opinion in Environmental Sustainability*, 58, 101217.
- Mechler, R., Bouwer, L. M., Linnerooth-Bayer, J., Hochrainer-Stigler, S., Aerts, J. C., Surminski, S., & Williges, K. (2014). Managing unnatural disaster risk from climate extremes. *Nature Climate Change*, 4(4), 235-237.
- Van Vliet, M. T., Wiberg, D., Leduc, S., & Riahi, K. (2016). Power-generation system vulnerability and adaptation to changes in climate and water resources. *Nature Climate Change*, 6(4), 375-380.